# Predicting Out-of-Domain Generalization with Neighborhood Invariance

**Nathan Ng**                                                       *nathanng@mit.edu*
*University of Toronto*
*Vector Institute*
*Massachusetts Institute of Technology*

**Neha Hulkund**                                                    *nhulkund@mit.edu*
*Massachusetts Institute of Technology*

**Kyunghyun Cho**                                             *kyunghyun.cho@nyu.edu*
*New York University*
*Prescient Design, Genentech*
*CIFAR Fellow*

**Marzyeh Ghassemi**                                              *mghassem@mit.edu*
*Massachusetts Institute of Technology*
*CIFAR AI Chair*
*Vector Institute*

**Reviewed on OpenReview:** *https://openreview.net/forum?id=jYkWdJzTwn*

## Abstract

Developing and deploying machine learning models safely depends on the ability to characterize and compare their abilities to generalize to new environments. Although recent work has proposed a variety of methods that can directly predict or theoretically bound the generalization capacity of a model, they rely on strong assumptions such as matching train/test distributions and access to model gradients. In order to characterize generalization when these assumptions are not satisfied, we propose neighborhood invariance, a measure of a classifier's output invariance in a local transformation neighborhood. Specifically, we sample a set of transformations and given an input test point, calculate the invariance as the largest fraction of transformed points classified into the same class. Crucially, our measure is simple to calculate, does not depend on the test point's true label, makes no assumptions about the data distribution or model, and can be applied even in out-of-domain (OOD) settings where existing methods cannot, requiring only selecting a set of appropriate data transformations. In experiments on robustness benchmarks in image classification, sentiment analysis, and natural language inference, we demonstrate a strong and robust correlation between our neighborhood invariance measure and actual OOD generalization on over 4,600 models evaluated on over 100 unique train/test domain pairs.

## 1 Introduction

As deep neural networks find increasing use in safety-critical domains such as autonomous driving (Gupta et al., 2021) and healthcare (Wiens et al., 2019), it is important to develop methods to understand and compare how these models generalize to new environments. Although empirically these models generalize in many settings (Hendrycks et al., 2020a; Allen-Zhu et al., 2018; Neyshabur et al., 2017a) , they also exhibit numerous failure cases. For example, models have been shown to overfit to a dataset's meta characteristics (Recht et al., 2019) or arbitrarily corrupted labels (Zhang et al., 2016), learn spurious correlations (Liang

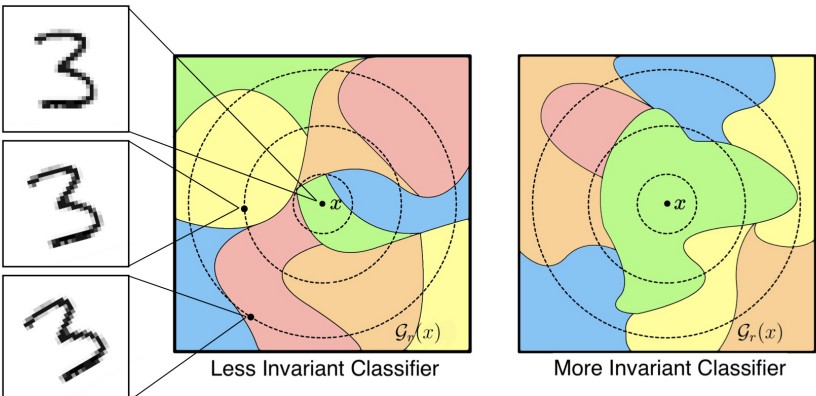

Figure 1: The transformation neighborhood $\mathcal{G}_r(x)$ around $x$ contains the set of points reachable from a transformation in $\mathcal{G}_r$ (pictured here as a rotation of $r$ degrees). As $r$ increases, the transformation neighborhood is partitioned into a distinct set of decision regions. More invariant classifiers (right) classify most points in the neighborhood into the same class and should generalize better compared to less invariant classifiers (left). Our measure of invariance is independent of what specific region $x$ lies in.

& Zou, 2022), and change their predictions even with small adversarial perturbations (Goodfellow et al., 2014; Papernot et al., 2017). Many methods have been proposed to mitigate these issues, but precisely characterizing the generalization properties of a model in diverse settings remains an open problem.

One line of work aims to theoretically bound generalization capacity (Vapnik & Chervonenkis, 1971; Bartlett & Mendelson, 2003; McAllester, 1999; Neyshabur et al., 2017a; Dziugaite & Roy, 2017; Neyshabur et al., 2015b) or directly predict generalization (Keskar et al., 2016; Liang et al., 2019; Neyshabur et al., 2015a; Schiff et al., 2021; Jiang et al., 2019), and are useful in reasoning about a model beyond its performance on a specific known test set. However, these methods work only when train and test distributions are the same, and often rely on a strong set of assumptions such as access to labelled test data (Schiff et al., 2021), model weights (Neyshabur et al., 2015b; Bartlett et al., 2017; Neyshabur et al., 2017b), model gradients (Jiang et al., 2019), and training data (Keskar et al., 2016). More recent work aims to estimate the generalization of a trained model on unlabelled test data directly (Deng & Zheng, 2021; Jiang et al., 2021; Deng et al., 2021; Garg et al., 2022). However, these metrics are typically calculated based on the output logits of a model on individual examples, which can become poorly calibrated in out-of-domain (OOD) settings (Morteza & Li, 2022). In real world settings, we require a robust measure of generalization that can be applied across a wide range of test distributions and where we are often given access only to a black box model.

In this paper we propose *neighborhood invariance*, a complexity measure that correlates well with generalization and that only assumes access to a set of suitable data transformations. Given a test data point, we define the transformation neighborhood as the set of points that can be generated from a set of transformations with a given maximum magnitude. A classifier's neighborhood invariance is then the proportion of points that are classified into the most commonly predicted class in this neighborhood. Intuitively, a classifier that is more invariant in this neighborhood should have be able to represent examples with lower dimensionality and thus lower complexity, leading to stronger generalization. Different from other similar methods (Aithal K et al., 2021), we define invariance with respect to the neighborhood itself rather than relative to the prediction at the test point and do not require manually tuning weights, meaning our measure can be applied even when test distributions vary. In addition, since our measure makes so few assumptions it is applicable in a wide range of experimental settings and can be used to compare the generalization properties of multiple models even when labeled data is unavailable.

We investigate the correlation of a model's neighborhood invariance with its capacity to generalize, focusing on experimental settings with OOD dataset shifts (Taori et al., 2020) where test data is sampled from a

distribution different from the training distribution. We select common OOD benchmark datasets in image classification (Krizhevsky, 2009; Lu et al., 2020; Recht et al., 2018; Deng, 2012; Darlow et al., 2018; Netzer et al., 2011; Arjovsky et al., 2019; Taori et al., 2020), sentiment analysis (Ni et al., 2019), and natural language inference (Williams et al., 2018), which totals over 100 pairs of training/test domains. We consider a large pool of over 4,600 models trained on these datasets with varying architectures and generalization properties, and sample sets of transformations commonly used for data augmentation (Ng et al., 2020; Cubuk et al., 2020; Wei & Zou, 2019; Xie et al., 2019). Across a wide set of correlation metrics, we find that neighborhood invariance measures outperform or match baselines in almost all experimental settings.

## 2 Related Work

**Characterizing Model Invariance**   Ensuring various kinds of model invariance is a well studied aspect of learning generalizable models and has been analyzed extensively from a causality perspective (Bühlmann, 2018; Peters et al., 2015; Haavelmo, 1943). At the largest scale, models trained on a wide support of training data and domains have demonstrated robust zero-shot and few-shot abilities (Radford et al., 2021; Brown et al., 2020; Wortsman et al., 2022). At a smaller scale, models that are invariant across data domains or interventions (Arjovsky et al., 2019; Gulrajani & Lopez-Paz, 2020; Bühlmann, 2018) are able to learn representations that do not depend on spurious correlations. Finally, at the smallest scale, local invariance to data augmentations (Cubuk et al., 2020), local changes (Rifai et al., 2011), augmentation graphs (HaoChen et al., 2021), similar neighbors (Luo et al., 2018), or interpolation between points (Verma et al., 2019; Zhang et al., 2018) have demonstrated improvements in model generalization. In our paper, we consider model invariance at this local scale.

Recent work has shown that models that are invariant to local transformations factorize the input space into a base space and the set of transformations (Sokolić et al., 2017; Sannai et al., 2021), effectively reducing the input dimensionality and thus model complexity (Anselmi et al., 2016; Anselmi et al., 2015). Measuring this decrease in complexity can be performed by analyzing the sample cover (Zhu et al., 2021). A similar line of work derives estimation error bounds based on the intrinsic dimensionality of deep ReLU networks in Hölder (Schmidt-Hieber, 2019; Nakada & Imaizumi, 2020; Chen et al., 2019), Besov, mixed smooth Besov (Suzuki, 2018), and anisotropic Besov (Suzuki & Nitanda, 2021) function spaces.

Most similar to our work, Aithal K et al. (2021) measures a model's robustness to perturbations as a proxy for generalization. Our method generalizes theirs and differs in a few key ways. We calculate our measure on the test set relative to a transformation neighborhood and can thus adapt to any specific domain for which we measure complexity and predict generalization. In contrast, Aithal K et al. (2021) calculate their measure on the training set, use the models' prediction as a ground truth, and require manually tuning the weights of augmentations, meaning it is relatively brittle and can only be applied to in-domain data. In addition we analyze the correlation of our neighborhood invariance measure on a wide range of OOD benchmarks on image classification, sentiment analysis, and natural language inference, while Aithal K et al. (2021) consider only image classification tasks with matching train/test distributions.

**Measures of Complexity and Predicting Generalization**   Traditional methods of analyzing the generalization bounds of neural networks use theoretical measures of complexity. VC dimension (Vapnik & Chervonenkis, 1971) and Rademacher complexity (Bartlett & Mendelson, 2003) can be used to bound the generalization of particular function classes, although they are often vacuous at the scale of deep neural networks (Dziugaite & Roy, 2017). The PAC-Bayes framework (McAllester, 1999; Neyshabur et al., 2017a; Dziugaite & Roy, 2017; Garg et al., 2021) can be used to build tighter generalization bounds by considering the "sharpness" of the local minima. Norm-based measures (Neyshabur et al., 2015b; Bartlett et al., 2017; Neyshabur et al., 2017b) bound generalization by considering different norms of the weights of learned networks. More recent analyses have focused on empirically motivated measures that do not provide theoretical bounds. These include the sharpness of minima in parameter space Keskar et al. (2016), Fisher-Rao norm Liang et al. (2019), distance from initialization (Nagarajan & Kolter, 2019), path norm (Neyshabur et al., 2015a), layer margin distributions (Jiang et al., 2019), and perturbation response curves Schiff et al. (2021).

However, these measures are only applicable when train and test distributions match. Although some generalization bounds have been derived for these OOD settings (Garg et al., 2021; Ben-David et al., 2007;

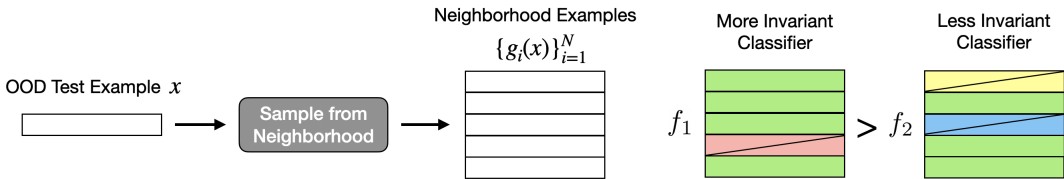

Figure 2: To estimate neighborhood invariance we sample a set of transformations $\{g_i\}_{i=1}^N \sim \mathcal{G}_r$ and generate a set of nearby examples $\{g_i(x)\}_{i=1}^N$ for every test example $x$. This set of examples is then evaluated using each classifier. We expect classifiers with more points classified in the most common class to generalize better to the given test set.

Zhang et al., 2019), they rely on access to the test data distribution. In addition, many testbeds examine only synthetic shifts, whereas natural shifts such as WILDS (Koh et al., 2021) are much more difficult. In real world settings where test distributions are often unknown, a separate line of work aims to directly predict generalization from unlabelled test data. These methods either predict the correctness on individual examples (Deng & Zheng, 2021; Jiang et al., 2021; Deng et al., 2021), directly estimate the total error (Garg et al., 2022; Guillory et al., 2021; Chen* et al., 2021; Chuang et al., 2020; Vedantam et al., 2021), or learn linear models relating ID and OOD accuracy (Miller et al., 2021) or agreement (Baek et al., 2022).

## 3 Neighborhood Invariance Measure

In this section we introduce our neighborhood invariance measure. We start by defining the transformation neighborhood of a point, then motivate our formulation of invariance in this neighborhood, and finally show how to estimate it in practice.

### 3.1 Motivation

Consider a classification task from an input space $\mathcal{X}$ to an output space $\mathcal{Y}$ with $k$ classes. We are given a model $f : \mathcal{X} \to \mathcal{Y}$ trained on an in-domain training dataset $\mathcal{D}_i = \{(x_i^1, y_i^1), \ldots, (x_i^n, y_i^n)\}$ sampled from a distribution $P_i(\mathcal{X}, \mathcal{Y})$, and an out-of-domain test dataset $\mathcal{D}_o = \{(x_o^1, y_o^1), \ldots, (x_o^m, y_o^m)\}$ sampled from a distribution $P_o(\mathcal{X}, \mathcal{Y})$. We assume further that domains are covariate shifted such that $P(\mathcal{Y}|\mathcal{X})$ does not change between domains

We consider a set of data transformations $\mathcal{G}_r = \{g : \mathcal{X} \to \mathcal{X} \mid m(g) < r\}$ where $g$ is a particular data transformation with an associated measure of the magnitude of the transformation $m(g)$. For example, a set of image rotation transformations with a maximum angle of 30 degrees might be denoted $\mathcal{G}_{30}$ where $m(g_i) = \alpha$ is the angle of rotation for a specific $g_i$. For a given test point $x \in \mathcal{D}_o$, we define the transformation neighborhood $\mathcal{G}_r(x) = \{g(x) \in \mathcal{X} \mid g \in \mathcal{G}_r\}$ as the set of outputs after applying all transformations in $\mathcal{G}_r$. Defining the neighborhood this way allows us to consider a wide range of nearby points in a controllable way without needing access to the underlying data distribution. As shown in Figure 1, for a given $r$, we can define a neighborhood decision distribution as

$$p_j(x) = \frac{|\{f(x') = j \mid x' \in \mathcal{G}_r(x)\}|}{|\mathcal{G}_r(x)|}. \tag{1}$$

We then define our **neighborhood invariance measure** as

$$\mu(f, x) = \max_{j \in \mathcal{Y}} p_j(x). \tag{2}$$

We assume that data transformations are selected such that the label for the transformed point $g(x)$ is still well defined. For example, flipping MNIST digits horizontally would cause most examples to have an undefined label. If the label is undefined, then $f(g(x))$ should produce close to random outputs and thus

constant invariance values of $1/k$ regardless of the generalization properties of $f$, causing our measure to fail. We empirically measure this phenomenon across a wide range of transformations in Section 5.4.

Intuitively, a classifier that is more invariant in the neighborhood of $x$ should be able to represent it with a lower input dimensionality and thus be less complex, leading to stronger generalization capabilities. In contrast, a less invariant classifier will need a higher input dimensionality to represent the neighborhood of $x$ and thus be more complex, leading to weaker generalization. Crucially, since our invariance is measured with respect to the neighborhood around the test point rather than the test point itself, it does not rely on the ground truth label. In addition, it makes no assumptions about the model or the distribution from which test data was sampled, making it applicable in many settings where existing complexity measures cannot be calculated, including common OOD robustness settings.

### 3.2 Estimating Neighborhood Invariance

Calculating neighborhood invariance exactly is typically intractable since evaluating all possible transformations is impossible. Instead, we perform Monte Carlo estimation by sampling a set of $N$ transformations $\{g_i\}_{i=1}^N$ from $\mathcal{G}_r$ (including the identity transformation $I(x) = x$) and calculating

$$\mu(f, x) = \frac{1}{N} \sum_{i=1}^N \mathbb{1}\left(f(g_i(x)) = \hat{y}(f, x)\right), \tag{3}$$

where

$$\hat{y}(f, x) = \arg\max_{j \in \mathcal{Y}} \sum_{i=1}^N \mathbb{1}\left(f(g_i(x)) = j\right) \tag{4}$$

is the most commonly output label. The average neighborhood invariance across the entire dataset $\mathcal{D}_o$ is then $\frac{1}{m} \sum_{j=1}^m \mu(f, x_o^j)$.

## 4 Experimental Setup

Empirically evaluating the quality of a complexity measure is difficult and requires careful experimental design. Typically, evaluation is done by generating a large pool of models with sufficiently varied generalization properties, but if we generate these models by varying only a few hyperparameters, our observed correlation may be an artifact of these factors affecting both generalization and our measure. To this end, we follow a similar experimental setup to Jiang et al. (2019).

### 4.1 Data

For our experiments we focus on three tasks: large and small scale image classification, sentiment analysis on single sentences and natural language inference on sentence pairs. For each task we construct a set of datasets sampled from different data domains.

**Image Classification**    For image classification we begin by considering 7 datasets domain shifted from ImageNet (Deng et al., 2009; Russakovsky et al., 2015) These include ImageNetV2 (Recht et al., 2019) and Imagenet-Sketch (Wang et al., 2019) with the same output classes, as well as ObjectNet (Barbu et al., 2019), ImageNetVid, YTBB anchors (Gu et al., 2019; Recht et al., 2019), ImageNet-A (Hendrycks et al., 2021b), and ImageNet-R (Hendrycks et al., 2021a) with a smaller subset of output classes.

In addition to ImageNet datasets we construct two sets of smaller scale datasets. The first we call **CI10** and consists of CIFAR10 (Krizhevsky, 2009), CINIC10 (Darlow et al., 2018), CIFAR10.1(Recht et al., 2018), and CIFAR10.2(Lu et al., 2020). The second we call **Numbers** and consists of SVHN (Netzer et al., 2011), MNIST (Deng, 2012), and Colored MNIST (Arjovsky et al., 2019). Domains in each set share the same set of output classes.

**Sentiment Analysis (SA)**    We use the datasets subsampled from the Amazon reviews dataset (Ni et al., 2019) which contains product reviews from Amazon. Following Hendrycks et al. (2020b) and Ng et al. (2020),

| Model | Dataset | Training Domain | Batch Size | Depth | Width | Dropout | Weight Decay | Label Noise | Learning Rate | Batch Norm | Seed | Data Aug | # Converged | # Evaluations |
|---|---|---|---|---|---|---|---|---|---|---|---|---|---|---|
| CNN | Amazon | 10 | 3 | 3 | 3 | 3 | 3 | — | — | — | — | — | 2,418 | 24,180 |
| | MNLI | 5 | 3 | 3 | 3 | 2 | — | 3 | — | — | — | — | 796 | 39,800 |
| RoBERTa | Amazon | 10 | 3 | — | — | 2 | 3 | 3 | — | — | — | — | 332 | 33,200 |
| | MNLI | 5 | 3 | — | — | 2 | 3 | 3 | — | — | — | — | 213 | 10,650 |
| Various | ImageNet | — | — | — | — | — | — | — | — | — | — | — | 401 | 2,406 |
| NiN | SVHN | — | 3 | 3 | — | 3 | 2 | — | — | — | — | — | 54 | 162 |
| | CIFAR10 | — | 2 | 2 | 2 | 2 | 2 | — | — | — | — | — | 32 | 128 |
| VGG | CIFAR10 | — | 3 | 3 | — | 3 | 2 | — | — | — | — | — | 54 | 216 |
| ResNet | CIFAR10 | — | — | — | 3 | — | 3 | — | 2 | 2 | 3 | 2 | 216 | 864 |
| CNN | CINIC10 | — | 2 | 2 | 4 | — | 2 | — | 2 | 2 | — | — | 128 | 512 |
| | | | | | | | | | | | | | 4,644 | 112,118 |

Table 1: Number of possible hyperparameter values for each architecture and task. Fields denoted with a — indicate that this hyperparameter is fixed or not applicable. We also list the total number of models converged and evaluations run in each model pool. In total we consider 4,644 models and 112,118 evaluations.

we split the dataset into 10 different domains based on review category. For all domains and datasets, models are trained to predict a review's star rating from 1 to 5.

**Natural Language Inference (NLI)**    We use the MNLI (Williams et al., 2018) dataset, a corpus of NLI data from 10 distinct genres of written and spoken English. We train on the 5 genres with training data and evaluate on all 10 genres. Models are given two sentences, a premise and hypothesis, and predict whether the hypothesis is entailed by, is neutral to, or contradicts the premise.

## 4.2   Model and Hyperparameter Space

For large scale image classification on ImageNet, we use pretrained models from the ImageNet Testbed (Taori et al., 2020) which covers a wide range of architectures including ResNext (Xie et al., 2016), EfficientNet (Tan & Le, 2019), BiT (Beyer et al., 2021), Vision Transformers (Dosovitskiy et al., 2020), CLIP (Radford et al., 2021), and many more models. We provide a full list of models evaluated in Appendix A.2. For smaller scale image classification tasks, we use models trained for the tasks 1, 2, 4, 5, and 9 from the Predicting Generalization in Deep Learning competition (PGDL) (Jiang et al., 2020) as well as models from Jiang et al. (2019), which covers Network in Network (NiN) (Lin et al., 2013), VGG (Simonyan & Zisserman, 2015), ResNet (He et al., 2015), and CNN models trained on CIFAR10, CINIC10, and SVHN. On natural language tasks we consider CNN (Kim, 2014; Mou et al., 2016) and RoBERTa (Liu et al., 2019) based models. On natural language models we apply label noise by randomly replacing a fraction of training labels with uniform samples from the label space. We argue that label noise is not an artificial training setting as stated in Jiang et al. (2019) but rather a method of entropy regularization (Pereyra et al., 2017; Xie et al., 2016) which prevents models from becoming overconfident.

In order to control for the varying convergence rates and learning capacities of our different models, we follow Jiang et al. (2019) and early stop the training of models when they reach a given training cross entropy loss (usually around 99% training accuracy), or if they reach the max number of training epochs. We discard all models which do not converge within this time. The total number of models trained and converged in each pool as well as details on hyperparameter variations for each task and model provided in Table 1. We include further details on model training, the hyperparameter space, and specific choices in hyperparameters in Appendix A.4, A.2, and A.3.

## 4.3   Evaluation Metrics

Given a set of domains defined by distributions $\{P_1, P_2, \ldots P_n\}$ and a set of datasets $\{\mathcal{D}_i \sim P_i\}_{i=1}^n$ sampled from these domains, we train a set of models $F_i = \{f_i^1, f_i^2, \ldots, f_i^m\}$ on each dataset $\mathcal{D}_i$. We evaluate all models $f_i^k \in F_i$ on all OOD test datasets $\mathcal{D}_o : o \neq i$, generating a set of invariance and generalization values $(\mu_{io}^k, g_{io}^k)$. We define generalization as the top-1 accuracy of $f_i^k$ on $\mathcal{D}_o$.

We evaluate our measure first by predicting the generalization of a given model to an OOD test set. Specifically, we select an OOD test set $\mathcal{D}_o$ and an in-domain training set $\mathcal{D}_i : i \neq o$ and predict the OOD generalization $g_{io}^k$

of a model $f_i^k$ trained on $\mathcal{D}_i$ and evaluated on $\mathcal{D}_o$ from its invaraince value $\mu_{io}^k$. To generate these predictions we use a linear model $\hat{g} = a\mu + b$ with parameters $a, b \in \mathcal{R}$. To estimate our parameters $a$ and $b$, we select a pool of models $\{f_j^k \in F_j : j \neq i, o\}$ that are trained on *all remaining datasets*. Each model $f_j^k$ in this pool is evaluated on the OOD dataset $\mathcal{D}_o$ to give us a set of pairs $\{(\mu_{jo}^k, g_{jo}^k)\}$. We then find $a, b$ by minimizing the mean squared error $(a^*, b^*) = \arg\min_{a,b} \sum_{j,k} (a\mu_{jo}^k + b - g_{jo}^k)^2$ on all models in the pool.

We use the learned parameters to make generalization predictions $\hat{g}_{io}^k = a\mu_{io}^k + b$ for every model $f_i^k \in F_i$ on $\mathcal{D}_o$ and measure the coefficient of determination $\mathbf{R^2}$ (Glantz et al., 1990). We also measure the residuals of our linear model by calculating the mean absolute error (**MAE**) between our predictions and the actual generalization. For every pair of training domain $i$ and OOD test domain $o$, we evaluate $R^2$ and MAE then average each metric across all pairs. We report MAE values as percentage points.

We also consider the rank correlation between neighborhood invariance and actual generalization. Specifically, for a pair of models $f_i, f_j$ with measure and generalization pairs $(\mu_i, g_i)$ and $(\mu_j, g_j)$, we want $g_i > g_j$ if $\mu_i > \mu_j$. We use Kendall's rank $\tau$ coefficient (Kendall, 1938) to measure how consistent these sets of rankings are. We measure four different $\tau$ values:

**ID $\tau$**    This metric evaluates the correlation of our measure with in-domain generalization. We select a training dataset $\mathcal{D}_i$ and consider pairs $\{(\mu_{ii}^k, g_{ii}^k)\}$ generated from the set of models $F_i$ trained on $\mathcal{D}_i$. $\tau$ values are averaged across all training domains.

**Macro $\tau$**    This metric evaluates the correlation of our measure individually on each training/OOD test domain pair. We select a training dataset $\mathcal{D}_i$ and a OOD test dataset $\mathcal{D}_o$ and consider pairs $\{(\mu_{io}^k, g_{io}^k)\}$ generated from the set of models $F_i$ trained on $\mathcal{D}_i$. $\tau$ values are averaged across all pairs of training and OOD test domains.

**Micro $\tau$**    This metric evaluates the correlation of our measure on a given OOD test domain across models trained on all other domains. We select a single OOD test domain $\mathcal{D}_o$ and consider pairs $\{(\mu_{io}^k, g_{io}^k)\}$ generated from the set of models $\{f_i^k \in F_i : i \neq o\}$ trained on all other datasets $\{D_i : i \neq o\}$. $\tau$ values are averaged across all test domains. We use this metric only when different models are trained on different training sets.

**Arch $\tau$**    This metric evaluates the correlation of our measure on models trained with different architectures. Arch $\tau$ is calculated similar to Micro $\tau$, except $F_i$ now includes models from all architectures. $\tau$ values are averaged across all test domains.

### 4.4    Data Transformations

Defining the transformation neighborhood requires defining a set of data transformations with associated magnitudes. For image classification, we consider four transformations: RandAugment (Cubuk et al., 2020) which randomly combines various transformations, random translation in the X- and Y-axes, random patch erasing (Zhong et al., 2020) which removes randomly sized patches from the image, and horizontal flips and crops. We call neighborhood invariance measures based on these transformations **NI-RandAug**, **NI-Translate**, **NI-Erase**, and **NI-FC** respectively. For natural language tasks, we also consider four transformations: SSMBA (Ng et al., 2020) which generates examples in a manifold neighborhood using a denoising autoencoder, EDA (Wei & Zou, 2019) which applies random word level operations, backtranslation (BT) (Rico Sennrich, 2016; Xie et al., 2019) which translates back and forth from a pivot language, and a transformation that randomly replaces a percentage of tokens. We call neighborhood invariance measures based on these transformations **NI-SSMBA**, **NI-EDA**, **NI-BT**, and **NI-RandRep** respectively. For all experiments we sample $n = 10$ transformations in addition to the identity tranfsormation, although ablations in section 5.4 show that our method is relatively robust to the specific number of transformations sampled. We provide further details on specific transformation magnitude values and implementations for all methods in Appendix A.5.

### 4.5    Baselines

Since our experimental setting makes so few assumptions, there are very few complexity measures that we can compare against. This includes Aithal K et al. (2021), which requires matching train/test distributions.

We thus consider complexity measures that require only model weights, specifically the **Spectral** (Yoshida & Miyato, 2017; Neyshabur et al., 2017b) and **Frobenius** (Neyshabur et al., 2015b) norms. However, in our experiments we find close to 0 or negative correlation for these measures, so we do not report their performance. We also compare our method against output based methods that directly predict OOD generalization. We use **ATC-MC** and **ATC-NE** Garg et al. (2022) as our two baselines, which calculate a threshold on in-domain validation data based on max confidence and negative entropy scores respectively. To calculate metrics on these methods we treat the generated accuracy predictions as a score. To calculate ID $\tau$ values we select a threshold value based on validation data then calculate predicted accuracy values on test data from the same domain. For ImageNet domain shift datasets where the output classes are a subset of the original 1,000 ImageNet classes, we do not recompute a subclassed ATC threshold as we do not assume prior knowledge of the OOD output classes.

## 5 Results

We now present the results of our experiments evaluating the quality of our neighborhood invariance measure. We begin by analyzing the effect of dataset distance on the correlation of neighborhood invariance with generalization in a toy setting. Our main set of results evaluate neighborhood invariance on OOD benchmarks in image classification, sentiment analysis, and natural language inference. Finally, we examine the correlation of our measure in extreme OOD settings and analyze the factors that affect the quality of our neighborhood invariance estimates.

### 5.1 Dataset Distance: Toy Analysis

In general, as with any complexity measure or generalization predictor, we expect neighborhood invariance to perform more poorly as we move farther from the training domain. In the worst case, if a classifier becomes a degenerate constant classifier in a far enough OOD domain, then neighborhood invariance reaches a constant maximum value of 1 while generalization becomes random. In order for our measure to work well, we assume that test domains are sufficiently close to training domains so that model predictions are non-constant. In this section we present an analysis of the effect of dataset distance on the quality of our measure in a toy setting.

We consider a binary classification task of points inside and outside a unit hypersphere in $\mathcal{X} = \mathcal{R}^n$, as shown in Figure 2. We define different data domains as univariate gaussian distributions $P(\mathcal{X})$, centered at a point $\mu$ on the hypersphere. The distance between two datasets $\mathcal{D}_1 \sim P_1(\mathcal{X})$ and $\mathcal{D}_2 \sim P_2(\mathcal{X})$ can then be measured as the distance along the hypersphere between $\mu_1$ and $\mu_2$ in radians. We generate a training dataset $\mathcal{D}_0$ by sampling points around the north pole of the hypersphere, then generate out-of-domain datasets $\mathcal{D}_j$ at varying distances from $\mathcal{D}_0$. Given a model trained on $\mathcal{D}_0$, we can calculate its generalization to $\mathcal{D}_j$, as well as its neighborhood invariance.

In our experiments, we consider a 16 dimensional hypersphere and sample 1000 points per data distribution for each dataset. The univariate gaussian distributions that we sample data points from have fixed variance 0.005, ensuring models cannot generalize fully across the entire hypersphere. We train 200 single hidden-layer MLPs with hidden dimension of 16 on the training dataset $\mathcal{D}_0$, each with a random level of label noise between 0-30% to ensure a wide range of generalization properties. We consider 40 different dataset distances, equally spaced along the hypersphere between opposite poles. For a given dataset distance, we select 5 points at random from the corresponding circumference and generate 5 datasets from univariate gaussians centered at these points. To measure neighborhood invariance for a given $x$, we sample 10 transformations from the set of transformations defined as a perturbation along the hypersphere of radius $||x||$ with a maximum distance $m(g) = ||g(x) - x|| \leq 0.01$. For each dataset we measure the neighborhood invariance and generalization for each of the 200 trained models and calculate the Kendall $\tau$ correlation between them. For a given dataset distance, the $\tau$ values are then averaged across all datasets. Results are presented in Figure 3b.

For datasets closest to the training dataset, the correlation between generalization and neighborhood invariance is high. However, as dataset distance increases, correlation decreases. At a distance of around $\pi/4$, correlation becomes nearly 0, and continues decreasing until the two values are negatively correlated on data sampled

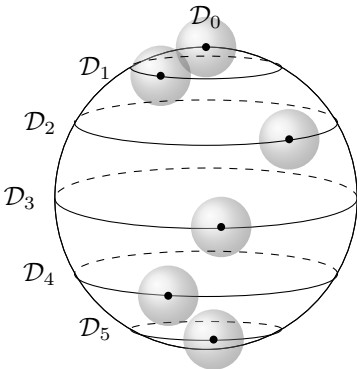

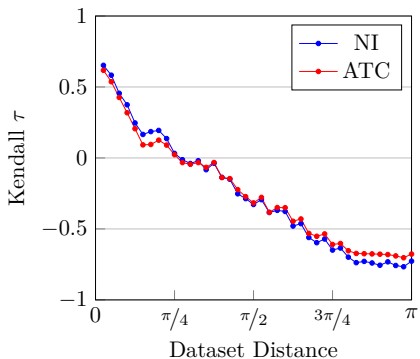

(a) We generate data from univariate gaussians whose means lie on a hypersphere. We train models on a dataset $\mathcal{D}_0$ to classify points as inside or outside the hypersphere then test them on out of domain datasets $\mathcal{D}_j$ that lie at various distances from $\mathcal{D}_0$ as measured by radiance distance along the hypersphere.

(b) The correlation between neighborhood invariance and OOD generalization remains high near the training domain but quickly decreases as dataset distance incerases, becoming negatively correlated for datasets further than $\pi/4$ away on the hypersphere. We observe almost identical results for baseline ATC methods.

Figure 3: Toy analysis of the effect of dataset distance on the correlation between neighborhood invariance and OOD generalization.

from the opposite side of the hypersphere from the training domain. This behavior almost identical for both neighborhood invariance and ATC methods (ATC-MC and ATC-NE perform the same so we report only one). These results demonstrate that the correlation of neighborhood invariance with generalization should decrease as dataset distance increases. To investigate the degree to which this happens in practice, we consider a set of extreme OOD experiments (Section 5.3) where we observe a surprisingly small decrease in correlation, indicating a closer dataset distance than might initially be assumed.

## 5.2 Correlation with OOD Generalization

We first present results in Table 2 analyzing the correlation of our proposed neighborhood invariance measure with OOD generalization. We report $R^2$, MAE, Macro $\tau$, Micro $\tau$, ID $\tau$, and Arch $\tau$ as detailed in Section 4.3. We omit results on Spectral and Frobenius norm measures as they are close to 0 or negative for all metrics. We do not report Micro $\tau$ values for image classification models since each model type is trained on only one domain. Additional experiments and results are presented in Appendix B.

**ImageNet-Scale Image Classification** Results on ImageNet-scale image classification datasets are presented in Table 28a and are averaged across all architectures. On standard domain shifts, NI-RandAug performs slightly better than ATC methods on $R^2$ and Macro $\tau$ with similar MAE. On the adversarial ImageNet-A dataset, ATC methods fail completely whereas NI methods maintain strong performance and still correlate well with accuracy. However, both methods exhibit large MAE and fail to accurately predict actual OOD accuracy. On ID $\tau$ NI methods perform slightly worse than ATC methods although they still show very strong correlations.

**CIFAR10-Scale Image Classification** Results on smaller scale image classification datasets are presented in Table 28b and are averaged across all architectures. On CI10 datasets, NI-RandAug significantly outperforms ATC baselines and all other measures on all metrics. NI-RandAug also exhibits only a small decrease in correlation when moving from in-domain (ID $\tau$) to OOD datasets (Macro $\tau$), compared to ATC methods which suffer a much larger drop. On Numbers datasets, NI-RandAug outperforms all other methods on $R^2$ and MAE, although it performs slightly worse on Macro $\tau$ compared to NI-Translate and on ID $\tau$ compared to ATC baselines.

For both CI10 and Numbers, using patch erasing and flip and crop transformations cause our method to perform worse or fail entirely, in contrast to ImageNet where they perform similarly or better. For these

| Measure | Domain Shifts | | | ImageNet-A | | | |
|---|---|---|---|---|---|---|---|
| | $R^2$ | MAE | Macro $\tau$ | $R^2$ | MAE | Macro $\tau$ | ID $\tau$ |
| NI-RandAug | **0.709** | 11.87 | **0.724** | 0.577 | **31.17** | 0.586 | 0.845 |
| NI-Translate | 0.587 | 13.58 | 0.604 | 0.468 | 29.54 | 0.439 | 0.834 |
| NI-Erase | 0.492 | 11.86 | 0.555 | 0.446 | 33.06 | 0.517 | 0.803 |
| NI-FC | 0.603 | 12.28 | 0.691 | **0.679** | 31.84 | **0.589** | 0.867 |
| ATC-NE | 0.607 | **11.40** | 0.703 | 0.209 | 32.00 | 0.248 | 0.931 |
| ATC-MC | 0.622 | 11.97 | 0.691 | 0.159 | 32.85 | 0.190 | **0.942** |

(a) Results on ImageNet scale models and datasets. On standard domain shift datasets NI-RandAug performs slightly better than ATC methods, and maintains strong performance on adversarial data where ATC methods fail completely.

| Measure | CI10 | | | | | Numbers | | | |
|---|---|---|---|---|---|---|---|---|---|
| | $R^2$ | MAE | Macro $\tau$ | ID $\tau$ | Arch $\tau$ | $R^2$ | MAE | Macro $\tau$ | ID $\tau$ |
| NI-RandAug | **0.899** | **3.11** | **0.768** | **0.793** | **0.837** | **0.764** | **5.33** | 0.642 | 0.733 |
| NI-Translate | 0.732 | 3.56 | 0.607 | 0.661 | 0.786 | 0.685 | 6.12 | 0.667 | **0.881** |
| NI-Erase | 0.518 | 4.41 | 0.411 | 0.406 | 0.299 | 0.153 | 10.17 | -0.135 | 0.324 |
| NI-FC | 0.417 | 4.47 | 0.371 | 0.344 | 0.683 | 0.208 | 10.42 | -0.316 | -0.033 |
| ATC-NE | 0.655 | 3.61 | 0.548 | 0.693 | 0.689 | 0.616 | 6.74 | 0.637 | 0.859 |
| ATC-MC | 0.640 | 3.65 | 0.544 | 0.682 | 0.685 | 0.692 | 6.19 | **0.682** | 0.844 |

(b) Results on small scale image classification, averaged across all model architectures. No Micro $\tau$ is reported since models are trained on a single domain and no Arch $\tau$ is reported for the Numbers dataset since we only consider a single architecture. NI-RandAug beats all other methods on almost all metrics.

| Measure | CNN | | | | | RoBERTa | | | | | |
|---|---|---|---|---|---|---|---|---|---|---|---|
| | $R^2$ | MAE | Macro $\tau$ | Micro $\tau$ | ID $\tau$ | $R^2$ | MAE | Macro $\tau$ | Micro $\tau$ | ID $\tau$ | Arch $\tau$ |
| NI-SSMBA | 0.662 | **1.93** | **0.677** | **0.689** | **0.629** | **0.972** | **1.29** | **0.832** | **0.829** | **0.838** | 0.588 |
| NI-EDA | 0.641 | 2.04 | 0.664 | 0.649 | 0.611 | 0.968 | 1.45 | 0.830 | 0.810 | 0.830 | 0.512 |
| NI-BT | 0.550 | 2.99 | 0.592 | 0.501 | 0.538 | 0.961 | 1.47 | 0.813 | 0.801 | 0.801 | 0.523 |
| NI-RandRep | 0.409 | 2.64 | 0.544 | 0.554 | 0.439 | 0.967 | 1.27 | 0.821 | 0.816 | 0.822 | 0.537 |
| ATC-NE | 0.760 | 2.47 | 0.514 | 0.633 | 0.467 | 0.852 | 2.38 | 0.707 | 0.691 | 0.749 | 0.660 |
| ATC-MC | **0.761** | 2.46 | 0.517 | 0.634 | 0.467 | 0.869 | 2.26 | 0.722 | 0.705 | 0.749 | **0.663** |

(c) Results on sentiment analysis (SA) datasets. NI-SSMBA beats all other methods on almost all metrics.

| Measure | CNN | | | | | RoBERTa | | | | | |
|---|---|---|---|---|---|---|---|---|---|---|---|
| | $R^2$ | MAE | Macro $\tau$ | Micro $\tau$ | ID $\tau$ | $R^2$ | MAE | Macro $\tau$ | Micro $\tau$ | ID $\tau$ | Arch $\tau$ |
| NI-SSMBA | 0.575 | 2.09 | 0.570 | **0.534** | 0.704 | 0.933 | 1.19 | 0.750 | 0.730 | 0.771 | 0.301 |
| NI-EDA | **0.577** | **2.04** | **0.581** | 0.511 | **0.709** | 0.941 | 1.26 | **0.789** | **0.757** | **0.799** | 0.572 |
| NI-BT | 0.509 | 2.11 | 0.470 | 0.449 | 0.584 | **0.944** | **1.07** | 0.759 | 0.740 | 0.778 | 0.563 |
| NI-RandRep | 0.451 | 2.20 | 0.452 | 0.428 | 0.570 | 0.890 | 1.70 | 0.688 | 0.647 | 0.710 | 0.401 |
| ATC-NE | 0.576 | 2.52 | 0.568 | 0.446 | 0.705 | 0.737 | 2.22 | 0.557 | 0.541 | 0.739 | 0.635 |
| ATC-MC | 0.576 | 2.52 | 0.568 | 0.446 | 0.706 | 0.769 | 2.10 | 0.581 | 0.567 | 0.748 | **0.636** |

(d) Results on natural language inference (NLI) datasets. NI measures beat baselines on all metrics except Arch $\tau$.

Table 2: Evaluation metrics measuring the correlation of our neighborhood invariance measure with ID/OOD generalization. The best performing measures for each metric are bolded. On all tasks, neighborhood invariance achieves strong generalization and beats baseline methods on almost all metrics. Full tables for $R^2$, Macro $\tau$, Micro $\tau$, and ID $\tau$ on individual train/test domains are in Appendix C

| Measure | CNN | | RoBERTa | |
|---|---|---|---|---|
| | $R^2$ | Micro $\tau$ | $R^2$ | Micro $\tau$ |
| NI-SSMBA | **0.584** | **0.566** | **0.941** | **0.816** |
| NI-EDA | 0.575 | **0.567** | 0.884 | 0.715 |
| NI-BT | 0.538 | 0.470 | 0.906 | 0.766 |
| NI-RandRep | 0.277 | 0.373 | 0.918 | 0.776 |
| ATC-NE | 0.271 | 0.495 | 0.329 | 0.356 |
| ATC-MC | 0.295 | 0.506 | 0.437 | 0.436 |

(a) Results on the `Drugs.com` dataset.

| Measure | CNN | | RoBERTa | |
|---|---|---|---|---|
| | $R^2$ | Micro $\tau$ | $R^2$ | Micro $\tau$ |
| NI-SSMBA | 0.083 | 0.080 | 0.691 | 0.463 |
| NI-EDA | 0.202 | 0.110 | **0.739** | **0.540** |
| NI-BT | **0.213** | **0.247** | 0.730 | 0.527 |
| NI-RandRep | 0.096 | 0.102 | 0.030 | 0.012 |
| ATC-NE | 0.077 | -0.107 | 0.719 | 0.345 |
| ATC-MC | 0.076 | -0.106 | 0.734 | 0.354 |

(b) Results on the MedNLI dataset.

Table 3: Evaluation metrics measuring the correlation of our neighborhood invariance measure with generalization on extreme OOD datasets. NI-* methods beat baselines on both tasks, with RoBERTa models exhibiting only a slight degradation in correlation compared to more typical OOD settings.

smaller datasets, since these transformations are more likely to generate images that cannot be classified (e.g. images without an object in frame) , they produce more similar invariance values between classifiers that cannot be used to rank them properly. This effect is more pronounced for Numbers datasets because flipping the image or removing even small portions of the number to be classified can render the task impossible. In contrast, random image translation which almost always preserves label information performs similarly well across both datasets and almost matches NI-RandAug. We provide a larger set of ablations on the Numbers dataset exploring this phenomenon in Section 5.4. The strong performance of NI-RandAug indicates that combining multiple transformations is helpful for mitigating dataset-specific transformation sensitivities, as in the case of Numbers.

**Sentiment Analysis (SA)**    Results on Sentiment Analysis datasets are presented in Table 28c. In experiments on both architectures, our neighborhood invariance measures achieves strong correlation with OOD generalization and beats all baselines on almost all metrics. Of the transformations considered, NI-SSMBA performs the best across both architectures. NI-EDA, NI-BT, and even NI-RandRep achieve strong results as well, often beating ATC baselines. We observe particularly strong correlation on RoBERTa models, with a nearly perfectly linear $R^2$ value of 0.972 and large Micro $\tau$ of 0.829. We hypothesize that this is due to the pretrained initialization of RoBERTa models, which gives a strong inductive bias towards learning a space invariant to transformations that preserve meaning. In contrast, CNN models are trained from random initializations and may not learn as closely aligned a space. On cross architecture analysis, we observe strong Arch $\tau$ for our neighborhood measures, although they are outperformed by both ATC methods. Compared to image classification results, our results on sentiment analysis tasks are overall less sensitive to the data transformations selected because they are less likely to destroy information necessary for classification.

**Natural Language Inference (NLI)**    Results on Natural Language Inference tasks are presented in Table 28d. Similar to our sentiment analysis results, our neighborhood invariance measures achieve strong correlation with OOD generalization on both architectures and beat all baselines. Correlations in general on NLI are lower than those of sentiment analysis because it is more difficult to maintain the complex relationship between the two sentences during a transformation. For example, changing a single word can easily change a sentence pair from entailment to contradiction, whereas many words must be changed to modify a 5 star review to a 1 star review. We observe exceptionally high correlation on RoBERTa models, for which we offer a similar hypothesis as in our sentiment analysis experiments. On cross architecture analysis, we observe strong correlation for our NI-EDA and NI-BT although they are outperformed by both ATC methods.

## 5.3   Extreme OOD Generalization

We now consider more extreme generalization to data domains with specialized and knowledge intensive data. We consider only natural language tasks as it is difficult to find a sufficiently specialized image classification dataset that maintains the same output classes. For sentiment analysis we use the `Drugs.com` review dataset (Gräßer et al., 2018), and for natural language inference we use MedNLI (Romanov & Shivade, 2018), an

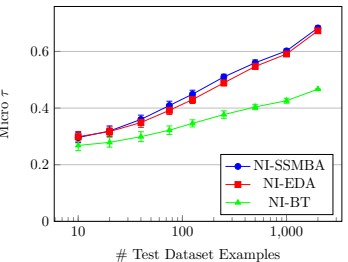
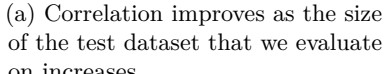

(a) Correlation improves as the size of the test dataset that we evaluate on increases.

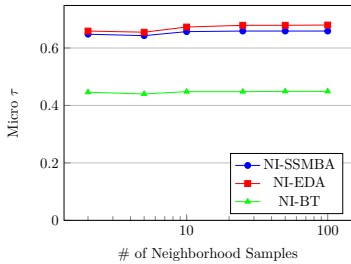

(b) Correlation is constant as the number of transformations decreases, even when we sample only a single transformation in addition to the identity transformation.

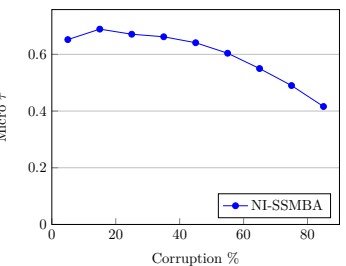

(c) Correlation increases until an optimal corruption percentage is reached, then decreases as the corruption continues to increase.

Figure 4: Micro $\tau$ for our neighborhood invariance measure calculated with varying ablations on CNN models evaluated on Amazon `toy` reviews.

NLI dataset generated from clinical notes and patient history. Both datasets contain highly specific medical language not seen in any of our training domains. All models from all original training domains are evaluated on each of these extreme OOD domains, and we report $R^2$ and Micro $\tau$. Results are shown in Table 3

On the `Drugs.com` dataset, we observe a small decrease in correlation for neighborhood invariance methods compared to results on AWS datasets. However, ATC methods begin to fail, with Micro $\tau$ on RoBERTa models dropping significantly from 0.706 to 0.356. This suggests that models become poorly calibrated in extreme OOD settings, making ATC methods fragile. On MedNLI we observe a much larger disparity in performance. For CNN models, most of our measures fail to correlate at all, and ATC methods degrade so much they became anti correlated with generalization. For RoBERTa models we observe only minor drops in correlation for all measures. For both tasks NI-RandRep exhibits almost no correlation with OOD generalization. This suggests that the choice of transformation becomes much more important as we move farther from our training domain.

## 5.4 Ablations

In this section we examine factors that may affect the quality of neighborhood invariance estimation and its correlation with actual generalization. Since rerunning all of our experiments is too costly, we evaluate on `toys` Amazon reviews using a pool of CNN models trained on all other domains for the first three ablations, and on the Numbers datasets using NiN models trained on SVHN for the final two.

**Test Dataset Size:** Does our neighborhood invariance measure still correlate well when the test dataset is small? We randomly and iteratively subsample our test dataset of 2000 examples to reduce our dataset size down to 10 examples. We then measure our models' neighborhood invariance on each subsampled dataset and calculate the Micro $\tau$ on all models. Results are shown in Figure 4a. We find that for all neighborhoods, smaller datasets lead to noisier invariance estimates and lower correlation. As dataset size increases, correlation increases as well.

**Number of Transformations:** How many transformations do we need to sample in order to generate a reliable neighborhood invariance estimate? We sample a varying number of transformations for each test example, from a minimum of two transformations to a maximum of 100 transformations, then estimate our neighborhood invariance measure with each set of transformations on the entire test dataset and calculate the micro $\tau$. By default we always include the identity transformation. Results are shown in Figure 4b. We find that our measure is surprisingly robust to the number of samples, with only a small difference between 100 and 2 transformations sampled. For all measures, correlation slightly increases as the number of samples increases and we achieve a better estimation of the true invariance value.

| Training Regime | $R^2$ | MAE | Macro $\tau$ | ID $\tau$ |
|---|---|---|---|---|
| Standard Training | 0.684 | 12.37 | **0.763** | 0.852 |
| + Augmentation/Robustness | 0.707 | 12.23 | 0.650 | **0.855** |
| + Pretraining/Extra Data | **0.799** | **11.73** | 0.707 | 0.836 |
| All Models | 0.709 | 11.87 | 0.724 | 0.845 |

Table 4: NI-RandAug results on different subsets of the ImageNet Testbed models trained with different training regimes. Training with augmentation and robustness interventions decreases macro $\tau$ compared to standard training but increases all other metrics. $R^2$ and MAE are higher for models trained/pretrained with large amounts of data.

**Transformation Magnitude:** How sensitive is our measure to the maximum magnitude of transformation considered? We use the SSMBA transformation for which the magnitude of a transformation is defined by the percentage of tokens corrupted, which we vary from a minimum of 5% to a maximum of 85%. After sampling a set of transformations from each corruption level, we estimate neighborhood invariance on the test dataset using each set and calculate the micro $\tau$ on all models. Results are shown in Figure 4c. We find that as we begin to increase the corruption percentage, correlation begins to increase as well. Correlation reaches a maximum, then decreases as we continue to increase our corruption percentage. However, even at 85% corruption, our method is quite robust and achieves a micro $\tau$ of 0.416.

**Selecting Transformations:** How do we ensure that transformations are suitable for a given dataset and do not destroy label information? We consider the set of NiN models trained on SVHN and a set of image transformations including RandAugment (Cubuk et al., 2020), rotations, translations, shears, brightness jittering, contrast jittering, color jittering, patch erasing, and flips and crops. For each transformation and both OOD datasets ColoredMNIST and MNIST we calculate the Macro $\tau$ correlation between accuracy and neighborhood invariance, as well as the average entropy difference between a model's output on a transformed image and on the original image. Results are shown in Figure 5.

Neighborhood invariance is relatively insensitive to the transformation selected, with most transformations performing similarly up to a certain entropy difference threshold around 0.1, after which it fails. The transformations that fail, erase and flip crop, both tend to destroy label information and lead to much higher entropy outputs. We propose this method of examining entropy differences as a simple way to diagnose whether a given transform is appropriate for a specific dataset.

**Pretraining and Training with Augmentation:** Does self-supervised pretraining or training with data augmentations, which should make models more invariant to certain transformations, make our neighborhood invariance measure ineffective? We begin by examining our metrics on subsets of models from the ImageNet Testbed (Taori et al., 2020) split by models trained on standard ImageNet (81 models), models trained on ImageNet with augmentations and robustness interventions (74 models), and finally models trained with extra data or pretrained with self-supervised objectives (41 models). Results are shown in Table 4. We find that compared to evaluating only on standard training models, the addition of augmentations or pretraining slightly degrades Macro $\tau$, but improves $R^2$ and MAE. Compared to the overall results on all models, we do not observe any large decreases in performance.

Since the augmentations considered in the set of ImageNet Testbed models are not the same across models, we also consider models from PGDL (Jiang et al., 2020) trained with and without a single type of augmentation: flip crop. Calculating our evaluation metrics on each set of models allows us to isolate the effect of data augmentation. Results are shown in Table 5. We find that measuring neighborhood invariance using the same transformation (NI-FC) that models are trained with causes only a slight degradation compared to models trained without. When measuring invariance using other transformations (NI-RandAug, NI-Erase), evaluation metrics actually improve slightly.

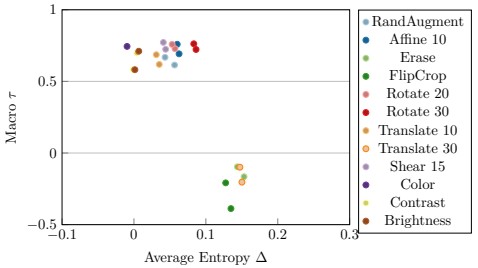

Figure 5: Transformations whose neighborhood invariance measures correlate well with OOD generalization exhibit smaller differences in output entropy.

| Measure | Flip Crop | $R^2$ | MAE | Macro $\tau$ | ID $\tau$ |
|---|---|---|---|---|---|
| NI-RandAug | ✗ | 0.833 | 3.32 | 0.719 | **0.753** |
| | ✓ | **0.844** | **3.16** | **0.732** | 0.744 |
| NI-Translate | ✗ | 0.874 | 2.61 | 0.768 | 0.831 |
| | ✓ | **0.877** | **2.52** | **0.794** | **0.845** |
| NI-Erase | ✗ | 0.790 | 3.28 | 0.716 | 0.702 |
| | ✓ | **0.812** | **3.17** | **0.722** | **0.719** |
| NI-FC | ✗ | **0.681** | 4.34 | **0.620** | **0.587** |
| | ✓ | 0.663 | **4.23** | 0.608 | 0.554 |

Table 5: Training with and without flip crop augmentation has a minimal effect on the effectiveness of our method, even when the transformation neighborhood aligns with those used to train the model (NI-FC).

## 6    Discussion

In this paper, motivated by the limited settings in which existing complexity measures can be applied, we propose a simple to calculate neighborhood invariance measure that can be applied even when test distributions are unknown and model training data, weights, and gradients are unavailable. We evaluate our method on image classification, sentiment analysis, and natural language inference datasets, calculating a variety of correlation metrics with both in-domain and out-of-domain (OOD) generalization. Across almost all tasks and experimental settings, we find that our neighborhood invariance measure consistently outperforms baseline methods and correlates strongly with actual generalization. However, our method has several limitations. Data transformations must be selected such that labels for transformed points are still well defined, although examining entropy differences can diagnose poor transformation choices In settings where such transformations are difficult to define, our method may not be applicable or provide inappropriately high estimates, so practitioners must be careful to verify their estimates with a labelled test set. In addition, our neighborhood invariance measure may fail in sufficiently OOD settings where a model may become poorly calibrated or degenerate, although we find in practice on our tasks that even extreme OOD settings are similar enough for our measure to perform well. In future work we plan to explore using similar measures calculated over transformation neighborhoods as a method for OOD detection.

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

# A    Experimental Setup

In this section we present full details of our experimental setup, including data preprocessing and specifics on model architecture and hyperparameter space. All models are trained on a single RTX6000 GPU.

## A.1 Data Preprocessing

Large ImageNet scale datasets are preprocessed using the pipeline provided by Taori et al. (2020). Small scale image classification datasets are preprocessed by normalizing pixel values and resizing to $32 \times 32$ if necessary. We use the same preprocessing steps for sentiment analysis and NLI experiments. All data is first tokenized using a GPT-2 style tokenizer and BPE vocabulary provided by `fairseq` (Ott et al., 2019). This BPE vocabulary consists of 50263 types. Corresponding labels are encoded using a label dictionary consisting of as many types as there are classes. Input text and labels are then binarized for model training.

## A.2 Model Architecture

The full list of the 196 models we evaluate from the Imagenet-Testbed (Taori et al., 2020) is provided below:

```
alexnet_lpf2, vit_large_patch32_384, wide_resnet101_2, resnet50_aws_baseline, resnet50_feature_cutmix, densenet169,
efficientnet-b2-autoaug, vgg11_bn, resnet50_with_jpeg_compression_aws, BiT-M-R101x3-nonfinetuned, efficientnet-b6-autoaug,
resnet18_lpf5, mobilenet_v2, resnet50_swsl, densenet121_lpf3, BiT-M-R101x1-nonfinetuned, efficientnet-b2-advprop-autoaug,
resnext101_32x8d_swsl, resnet50_with_fog_aws, resnet50_trained_on_SIN_and_IN, resnext101_32x4d, resnet50_with_-
contrast_aws, FixResNet50CutMix, resnet50_imagenet_subsample_1_of_32_batch64_original_images, resnext101_32x4d_-
swsl, squeezenet1_1, resnet50_imagenet_subsample_1_of_16_batch64_original_images, resnet18-rotation-nocrop_40,
resnet101_cutmix, efficientnet-b3, resnet50_with_motion_blur_aws, vit_large_patch16_384, efficientnet-b4, resnet50_-
lpf3, dpn107, resnext101_32x8d_deepaugment_augmix, vgg19, resnet18_ssl, vgg13_bn, vgg13, resnet50_with_pixelate_aws,
senet154, resnet18_lpf2, shufflenet_v2_x1_0, se_resnet101, alexnet_lpf5, densenet121, efficientnet-b3-advprop-autoaug,
resnet50_augmix, resnet50_simsiam, efficientnet-b0-advprop-autoaug, resnet50_imagenet_subsample_500_classes_batch64_-
original_images, vgg16_lpf2, mnasnet1_0, resnet34_lpf2, dpn68b, mobilenet_v2_lpf3, resnet101_lpf3, alexnet, vgg16_bn,
efficientnet-b0, inceptionv3, resnet18-rotation-worst10_30, resnet152_3x_simclrv2_finetuned_100pct_tf_port, resnet50_-
imagenet_subsample_1_of_2_batch64_original_images, wide_resnet50_2, polynet, efficientnet-b7-randaug, dpn131, vgg16_-
bn_lpf2, instagram-resnext101_32x16d, vgg16_bn_lpf5, resnet50_linf_eps8_robust, efficientnet-b1-advprop-autoaug,
inceptionv4, vit_b_32_clip_zeroshot, resnet18-rotation-worst10_40, resnet50_imagenet_100percent_batch64_original_-
images, resnet50_with_frost_aws, efficientnet-b3-autoaug, resnet50_imagenet_subsample_125_classes_batch64_original_-
images, efficientnet-b7-autoaug, resnet50_ssl, vgg16_lpf5, vit_base_patch16_224, resnet34_lpf5, resnet152, resnext50_-
32x4d, FixPNASNet, resnet50_with_saturate_aws, FixResNet50CutMix_v2, densenet121_lpf5, resnet50_imagenet_subsample_1_of_-
4_batch64_original_images, resnet18-rotation-random_40, resnet50_adv-train-free, resnet18_lpf3, BiT-M-R50x3-nonfinetuned,
efficientnet-b7-advprop-autoaug, resnet50_with_spatter_aws, resnet50_trained_on_SIN, resnet50_simclrv2_finetuned_100pct_tf_-
port, pnasnet5large, BiT-M-R50x3-ILSVRC2012, resnet50_imagenet_subsample_250_classes_batch64_original_images, efficientnet-b5,
resnet50_deepaugment, efficientnet-b5-randaug, resnet50_lpf2, se_resnext50_32x4d, resnet50_clip_zeroshot, resnext50_32x4d_-
swsl, BiT-M-R50x1-nonfinetuned, BiT-M-R101x3-ILSVRC2012, resnet50_imagenet_subsample_1_of_8_batch64_original_images, vit_-
large_patch16_224, efficientnet-b1-autoaug, efficientnet-b6-advprop-autoaug, efficientnet-b5-autoaug, resnet50_with_zoom_-
blur_aws, resnext50_32x4d_ssl, FixResNet50_v2, resnet50_lpf5, resnet101, resnet18_swsl, efficientnet-b2, squeezenet1_0,
resnet152-imagenet11k, resnet50_simclrv2_linear_probe_tf_port, alexnet_lpf3, bninception, efficientnet-b8-advprop-autoaug,
resnet50_linf_eps4_robust, FixResNet50, mnasnet0_5, resnet50_mixup, densenet121_lpf2, resnet18-rotation-standard_40, se_-
resnext101_32x4d, resnet18-rotation-random_30, efficientnet-b0-autoaug, efficientnet-b4-autoaug, vgg11, resnext101_32x8d,
BiT-M-R50x1-ILSVRC2012, resnet50, resnet50_with_gaussian_noise_contrast_motion_blur_jpeg_compression_aws, shufflenet_v2_x0_5,
dpn92, xception, resnet152_3x_simclrv2_linear_probe_tf_port, dpn98, bninception-imagenet21k, efficientnet-b5-advprop-autoaug,
resnext101_32x16d_ssl, vit_base_patch32_384, densenet201, inceptionresnetv2, cafferesnet101, instagram-resnext101_32x8d,
resnet34, FixResNet50_no_adaptation, resnext101_32x8d_ssl, resnet101_lpf5, mobilenet_v2_lpf5, instagram-resnext101_32x32d,
nasnetamobile, mobilenet_v2_lpf2, resnet101_lpf2, se_resnet50, dpn68, resnet50_with_brightness_aws, resnext101_64x4d,
resnext101_32x4d_ssl, vgg19_bn, fbresnet152, resnet50_deepaugment_augmix, se_resnet152, resnet50_cutout, resnet50_cutmix,
resnet50_l2_eps3_robust, efficientnet-b1, resnet50_with_defocus_blur_aws, BiT-M-R101x1-ILSVRC2012, vgg16_bn_lpf3, resnet50_-
trained_on_SIN_and_IN_then_finetuned_on_IN, nasnetalarge, resnet50_with_gaussian_noise_aws, vit_base_patch16_384, resnet50_-
swav, resnet50_with_greyscale_aws, vgg16, resnet34_lpf3, efficientnet-b4-advprop-autoaug, vgg16_lpf3, resnet18, densenet161
```

Our small image classification models are Network in Network (NiN) (Lin et al., 2013), VGG (Simonyan & Zisserman, 2015), and CNN models. Training and hyperparemeter details for these models are provided in Jiang et al. (2020).

For natural language tasks, our CNN models are based on the architecture in Kim (2014). Our input embeddings are 512 dimensional, which we treat as our channel dimension. Our base model applies a set of three one dimensional convolutions of kernel size 3, 4, and 5 with 256 output channels. We modulate the number of stacked convolutions (depth) as well as the channel size (width). Each convolution generates a separate representation that is max pooled across the sequence and concatenated together. We feed this representation into a MLP classifier with a single hidden layer of 512 dimensions. We apply dropout of 0.2 to our inputs and MLP classifier.

Our RoBERTa models use a pre-trained RoBERTa$_{\text{BASE}}$ model provided by `fairseq`. Classification token embeddings are fed into an MLP classifier with a single hidden layer of 512 dimensions. All models are written within the `fairseq` framework (Ott et al., 2019) and trained on a single RTX6000 or T4 GPU.

| Hyperparameter | CNN | | RoBERTa | |
| | SA | NLI | SA | NLI |
|---|---|---|---|---|
| Batch Size | {32, 64, 128} | {32, 64, 128 } | {8, 16, 32} | {8, 16, 32} |
| Depth | {1, 2, 3 } | {1, 2, 3 } | 1 | 1 |
| Width | {128, 256, 512} | {128, 256, 512 } | 768 | 768 |
| Dropout | {0.0, 0.25, 0.5} | {0.0, 0.25} | {0.0, 0.1} | {0.0, 0.1} |
| Weight Decay | {0.0, 0.0001, 0.0005} | 0.0 | {0.0, 0.0001, 0.0005} | {0.0, 0.0001, 0.0005} |
| Label Noise | 0.0 | {0.0, 0.2, 0.4} | {0.0, 0.2, 0.4} | {0.0, 0.2, 0.4} |

Table 6: Possible hyperparameter values for each architecture and task.

### A.3 Model Hyperparameters

Hyperparameter values for image classification models are provided in Jiang et al. (2020). For natural language models we vary the following hyperparameters: training domain, batch size, depth, width, dropout, weight decay, and label noise. For training domains, on sentiment analysis we choose between `books, clothing, home, kindle, movies, pets, sports, tech, tools, toys`. For training domains on NLI, we choose between `slate, government, fiction, telephone, travel`. NLI datasets include additional test sets `oup, nineeleven, facetoface, verbatim, letters`. Possible values for all other hyperparameters are provided in Table 6

### A.4 Model Training

All models are trained with the Adam optimizer (Kingma & Ba, 2014) with $\beta = (0.9, 0.98)$ and $\epsilon = 1 \times 10^{-6}$. CNN models are trained with learning rate $1 \times 10^{-3}$ and RoBERTa models are trained with learning rate $1 \times 10^{-5}$. We use a inverse square root learning rate scheduler to anneal learning rate over training. We early stop CNN models on sentiment analysis at 0.04 cross entropy and on NLI at 0.03 cross entropy. We early stop RoBERTa models on sentiment analysis at 0.05 cross entropy and on NLI at 0.03 cross entropy. Training details for image classification models are provided in Jiang et al. (2020).

### A.5 Transformation Magnitudes

We define how to determine the magnitude of each data transformation below, and use a maximum value based on best practices provided in their respective papers.

- **RandAugment:** The magnitude of a transformation is determined by the magnitude parameter in the RandAugment algorithm as well as the number of augmentations applied. In our experiments we consider transformations with a maximum magnitude of 15, with 3 augmentations for larger ImageNet models and 1 augmentation for smaller models.

- **Translate:** The magnitude of a transformation is determined by the maximum percentage of the image the image will be translated in both the X- and Y-axes. In our experiments we consider translations of up to 10% of the size of the image in both axes.

- **Erase:** The magnitude of the erase transformation is determined by the percentage of the image erased. In our experiments we consider transformations that remove a maximum size of 33% of the total image area and an aspect ratio between 1/3 and 10/3.

- **Flip and Crop:** The magnitude of the flip and crop transformation is determined by the flip probability and the crop size. In our experiments we consider transformations that flip the image 50% of the time and crop the image with a lower bound of 8% of total image area and an aspect ratio between 3/4 and 4/3. Images are resized to $32 \times 32$ after cropping.

- **SSMBA:** The magnitude of a SSMBA transformation is determined by the percentage of tokens corrupted, where of the tokens selected, 10% are unmasked, 10% are randomly replaced, and the

remaining 80% are masked. In our experiments we consider SSMBA transformations with a maximum of 15% of tokens corrupted.

- **EDA:** The magnitude of an EDA transformation is determined by the percentage of tokens noised. In our experiments we consider transformations with a maximum of 10% of the tokens.

- **Backtranslation:** The magnitude of a backtranslation operation is determined by the temperature of the softmax-ed distribution from which we sample tokens. In our experiments we consider transformations with a maximum temperature of 0.7.

- **Random Replacement:** The magnitude of a random replacement operation is determined by the percentage of tokens replaced. In our experiments we consider transformations with a maximum of 15% of tokens replaced.

## B   Additional Experiments

### B.1   Norm-Based Complexity Measures

Following (Jiang et al., 2019), we calculate our spectral norm measure as $\Pi_{i=1}^{d}||\mathbf{W}_i||_2^2$ and Frobenius norm measure $\Pi_{i=1}^{d}||\mathbf{W}_i||_F^2$. We do not list results on these measures as the correlations are often negative or 0.

### B.2   Cross-Domain Correlation

In this set of experiments we measure the correlation between neighborhood invariance and generalization values of a single model trained on a single training domain evaluated across different OOD test domains. For natural language experiments we average correlations across all CNN and RoBERTa models and training domains and call these the **CNN** $\tau$ and **Roberta** $\tau$. Since these results are rank correlations over only 9 values, they are quite noisy. Results are presented in Table 7.

Neighborhood invariance performs quite poorly on both models, although they still outperforms ATC baselines. We hypothesize different regions of the input space may have different optimal levels of smoothness that achieve the lowest generalization error. Our value of interest is then not the absolute smoothness, but the *relative* smoothness compared to this optimal value. These values are the same when comparing different models evaluated on the same domain, but are not the same for the same model evaluated on different domains, making correlating across domains difficult. On the natural image manifold where domains are more well behaved and uniform compared to the natural language manifold, the relative smoothness may not differ much between domains allowing us to correlate our measure across domains.

### B.3   Negative Entropy Results

As an alternative to defining the invariance as the maximum value of the neighborhood decision distribution in Eq. 1, we also consider defining it using the negative entropy of the same distribution:

$$\mu(f, x) = \sum_{j \in \mathcal{Y}} p_j(x) \log p_j(x)$$

A full table of results including metrics calculated on neighborhood invariance measured with negative entropy is provided in Table 8. We refer to measures calculated with entropy as **NE-SSMBA**, **NE-EDA**, **NE-BT**, and **NE-Random**. For most metrics, NE-* methods perform similarly or slightly worse.

## C   Full Results

We provide a full breakdown of results on the correlation metrics $R^2$ (Tables 10, 11, 12, 13, 14, 15), macro $\tau$ (Tables 17, 18, 19, 20, 21, 22, micro $\tau$ (Tables 23, 24), and ID $\tau$ (Tables 25, 26, 27) for each set of datasets and models. We also provide additional standard deviations for all main results in Table 2.

| Task | Measure | CNN $\tau$ | RoBERTa $\tau$ |
|---|---|---|---|
| SA | NI-SSMBA | 0.360 | 0.010 |
| | NI-EDA | 0.431 | **0.266** |
| | NI-BT | 0.505 | 0.245 |
| | NI-RandRep | **0.570** | 0.150 |
| | ATC-NE | 0.543 | 0.228 |
| | ATC-MC | 0.539 | 0.224 |
| NLI | NI-SSMBA | 0.022 | 0.260 |
| | NI-EDA | 0.102 | 0.335 |
| | NI-BT | 0.089 | 0.333 |
| | NI-RandRep | **0.226** | **0.440** |
| | ATC-NE | 0.219 | 0.231 |
| | ATC-MC | 0.223 | 0.239 |

Table 7: Correlation metrics evaluating the ability of our smoothness measure to predict OOD generalization across test datasets. Our smoothness measures achieves strong correlation in image classification tasks but fails in natural language tasks.

| Task | Measure | CNN | | | | RoBERTa | | | |
|---|---|---|---|---|---|---|---|---|---|
| | | $R^2$ | MAE | Macro $\tau$ | Micro $\tau$ | $R^2$ | MAE | Macro $\tau$ | Micro $\tau$ |
| SA | NI-SSMBA | **0.662** | 1.93 | **0.677** | **0.689** | **0.972** | 1.29 | **0.832** | **0.829** |
| | NI-EDA | 0.641 | 2.04 | 0.664 | 0.649 | 0.968 | 1.45 | 0.830 | 0.810 |
| | NI-BT | 0.550 | 2.99 | 0.592 | 0.501 | 0.961 | 1.47 | 0.813 | 0.801 |
| | NI-RandRep | 0.409 | 2.64 | 0.544 | 0.554 | 0.967 | **1.27** | 0.821 | 0.816 |
| | NE-SSMBA | 0.595 | 2.53 | 0.708 | 0.713 | 0.971 | 1.32 | 0.830 | 0.824 |
| | NE-EDA | 0.534 | 2.71 | 0.698 | 0.674 | 0.965 | 1.55 | 0.825 | 0.801 |
| | NE-BT | 0.471 | 3.59 | 0.618 | 0.541 | 0.961 | 1.46 | 0.813 | 0.799 |
| | NE-RandRep | 0.283 | 3.37 | 0.570 | 0.552 | 0.964 | 1.34 | 0.818 | 0.809 |
| | ATC-NE | 0.530 | 3.80 | 0.506 | 0.642 | 0.849 | 3.59 | 0.684 | 0.706 |
| | ATC-MC | 0.528 | 3.76 | 0.507 | 0.642 | 0.863 | 3.54 | 0.698 | 0.716 |
| NLI | NI-SSMBA | 0.575 | 2.09 | 0.570 | **0.534** | 0.933 | 1.19 | 0.750 | 0.730 |
| | NI-EDA | 0.577 | **2.04** | 0.581 | 0.511 | 0.941 | 1.26 | **0.789** | **0.757** |
| | NI-BT | 0.509 | 2.11 | 0.470 | 0.449 | 0.944 | 1.07 | 0.759 | 0.740 |
| | NI-RandRep | 0.451 | 2.20 | 0.452 | 0.428 | 0.890 | 1.70 | 0.688 | 0.647 |
| | NE-SSMBA | 0.588 | 2.20 | 0.579 | 0.520 | 0.941 | 1.39 | 0.738 | 0.711 |
| | NE-EDA | **0.606** | 2.11 | **0.597** | 0.512 | 0.937 | 1.48 | 0.767 | 0.732 |
| | NE-BT | 0.536 | 2.27 | 0.480 | 0.422 | **0.954** | **1.12** | 0.764 | 0.750 |
| | NE-RandRep | 0.457 | 2.36 | 0.451 | 0.397 | 0.904 | 1.79 | 0.665 | 0.591 |
| | ATC-NE | 0.378 | 3.57 | 0.430 | 0.294 | 0.673 | 2.35 | 0.536 | 0.52 |
| | ATC-MC | 0.382 | 3.57 | 0.433 | 0.297 | 0.718 | 2.21 | 0.570 | 0.556 |

Table 8: Correlation metrics evaluating the quality of our neighborhood invariance measure on two tasks, sentiment analysis and natural language inference, and two architectures, CNN and RoBERTa. Details on metric calculations and baselines are provided in sections 4.3 and 4.5. This full table of results includes metrics calculated with neighborhood negative entropy measure as well.

| Measure | ImageNetV2 | ImagenNet-Sketch | ObjectNet | ImageNet-Vid | YTBB | ImageNet-R | ImageNet-A |
|---|---|---|---|---|---|---|---|
| NI-RandAug | 0.810 | 0.641 | **0.613** | **0.767** | **0.570** | **0.763** | 0.577 |
| NI-Translate | 0.794 | 0.436 | 0.461 | 0.642 | 0.395 | 0.560 | 0.468 |
| NI-Erase | 0.715 | 0.222 | 0.384 | 0.635 | 0.399 | 0.398 | 0.446 |
| NI-FC | 0.767 | 0.406 | 0.516 | 0.706 | 0.446 | 0.618 | **0.679** |
| ATC-MC | 0.980 | **0.710** | 0.451 | 0.552 | 0.200 | 0.463 | 0.159 |
| ATC-NE | **0.991** | 0.709 | 0.392 | 0.484 | 0.229 | 0.441 | 0.209 |

(Header spanning: Test Domain)

Table 9: Full $R^2$ metrics for all ImageNet test domains. We average values across models.

| Model | Train Domain | Measure | SVHN | Colored MNIST | MNIST |
|---|---|---|---|---|---|
| NiN | SVHN | NI-RandAug | — | **0.785** | 0.744 |
| | | NI-Translate | — | 0.728 | 0.642 |
| | | NI-Erase | — | 0.088 | 0.218 |
| | | NI-FC | — | 0.134 | 0.283 |
| | | ATC-NE | — | 0.506 | 0.725 |
| | | ATC-MC | — | 0.615 | **0.769** |

(Header spanning: Test Domain)

Table 10: Full $R^2$ metrics for all test domains for image classification models on SVHN, Colored MNIST, and MNIST.

| Model | Train Domain | Measure | CIFAR10 | CINIC10 | CIFAR10.1 | CIFAR10.2 |
|---|---|---|---|---|---|---|
| NiN | CIFAR10 | NI-RandAug | — | **0.898** | **0.927** | **0.876** |
| | | NI-Translate | — | 0.688 | 0.864 | 0.730 |
| | | NI-Erase | — | 0.404 | 0.526 | 0.547 |
| | | NI-FC | — | 0.109 | 0.028 | 0.000 |
| | | ATC-NE | — | 0.237 | 0.575 | 0.435 |
| | | ATC-MC | — | 0.252 | 0.538 | 0.390 |
| ResNet | CIFAR10 | NI-RandAug | — | 0.816 | 0.844 | **0.853** |
| | | NI-Translate | — | **0.889** | **0.899** | 0.836 |
| | | NI-Erase | — | 0.807 | 0.812 | 0.783 |
| | | NI-FC | — | 0.726 | 0.628 | 0.660 |
| | | ATC-NE | — | 0.736 | 0.782 | 0.693 |
| | | ATC-MC | — | 0.702 | 0.760 | 0.680 |
| VGG | CIFAR10 | NI-RandAug | — | **0.969** | **0.950** | **0.929** |
| | | NI-Translate | — | 0.849 | 0.844 | 0.809 |
| | | NI-Erase | — | 0.096 | 0.100 | 0.104 |
| | | NI-FC | — | 0.637 | 0.524 | 0.487 |
| | | ATC-NE | — | 0.557 | 0.774 | 0.724 |
| | | ATC-MC | — | 0.559 | 0.764 | 0.709 |
| CNN | CINIC10 | NI-RandAug | **0.922** | — | **0.876** | **0.865** |
| | | NI-Translate | 0.516 | — | 0.449 | 0.407 |
| | | NI-Erase | 0.603 | — | 0.504 | 0.548 |
| | | NI-FC | 0.416 | — | 0.397 | 0.395 |
| | | ATC-NE | 0.869 | — | 0.750 | 0.724 |
| | | ATC-MC | 0.868 | — | 0.741 | 0.716 |

(Header spanning: Test Domain)

Table 11: Full $R^2$ metrics for all test domains for image classification models on CIFAR10, CINIC10, CIFAR10.1, and CIFAR10.2.

| Train Domain | Measure | | Test Domain | | | | | | | | |
|---|---|---|---|---|---|---|---|---|---|---|---|
| | | books | clothing | home | kindle | movies | pets | sports | tech | tools | toys |
| books | NI-SSMBA | — | 0.688 | 0.771 | 0.752 | 0.683 | 0.823 | 0.762 | 0.697 | 0.711 | 0.729 |
| | NI-EDA | — | 0.720 | 0.725 | 0.699 | 0.689 | 0.810 | 0.693 | 0.663 | 0.727 | 0.805 |
| | NI-BT | — | 0.560 | 0.663 | 0.639 | 0.593 | 0.675 | 0.602 | 0.591 | 0.592 | 0.658 |
| | NI-RandRep | — | 0.413 | 0.418 | 0.331 | 0.343 | 0.479 | 0.373 | 0.304 | 0.401 | 0.496 |
| | ATC-NE | — | **0.765** | **0.859** | **0.828** | **0.777** | **0.834** | **0.834** | **0.791** | **0.814** | **0.872** |
| | ATC-MC | — | 0.759 | 0.854 | 0.821 | 0.771 | 0.828 | 0.830 | 0.790 | 0.808 | 0.871 |
| clothing | NI-SSMBA | **0.517** | — | 0.601 | **0.531** | **0.525** | **0.535** | 0.566 | **0.484** | 0.549 | 0.631 |
| | NI-EDA | 0.409 | — | 0.419 | 0.452 | 0.408 | 0.395 | 0.457 | 0.388 | 0.486 | 0.534 |
| | NI-BT | 0.234 | — | 0.164 | 0.267 | 0.273 | 0.121 | 0.191 | 0.164 | 0.193 | 0.355 |
| | NI-RandRep | 0.236 | — | 0.237 | 0.223 | 0.250 | 0.212 | 0.213 | 0.156 | 0.246 | 0.335 |
| | ATC-NE | 0.480 | — | 0.673 | 0.478 | 0.467 | 0.466 | 0.739 | 0.299 | **0.772** | 0.847 |
| | ATC-MC | 0.477 | — | **0.675** | 0.477 | 0.466 | 0.474 | **0.742** | 0.300 | **0.772** | **0.848** |
| home | NI-SSMBA | 0.533 | 0.545 | — | 0.524 | 0.511 | 0.665 | 0.648 | 0.576 | 0.604 | 0.592 |
| | NI-EDA | 0.412 | 0.554 | — | 0.451 | 0.433 | 0.586 | 0.496 | 0.417 | 0.525 | 0.657 |
| | NI-BT | 0.313 | 0.260 | — | 0.327 | 0.316 | 0.346 | 0.366 | 0.304 | 0.385 | 0.408 |
| | NI-RandRep | 0.273 | 0.260 | — | 0.236 | 0.325 | 0.286 | 0.299 | 0.270 | 0.282 | 0.370 |
| | ATC-NE | 0.608 | **0.861** | — | 0.675 | 0.618 | **0.838** | 0.838 | **0.720** | **0.863** | 0.894 |
| | ATC-MC | **0.612** | **0.861** | — | **0.679** | **0.626** | **0.838** | **0.839** | 0.719 | **0.863** | **0.895** |
| kindle | NI-SSMBA | 0.645 | 0.680 | 0.650 | — | 0.626 | 0.765 | 0.659 | 0.551 | 0.556 | 0.634 |
| | NI-EDA | 0.722 | **0.688** | 0.695 | — | 0.662 | 0.785 | 0.690 | 0.658 | 0.623 | 0.783 |
| | NI-BT | 0.655 | 0.674 | 0.684 | — | 0.625 | 0.745 | 0.704 | **0.695** | **0.649** | 0.735 |
| | NI-RandRep | 0.325 | 0.364 | 0.253 | — | 0.269 | 0.442 | 0.292 | 0.244 | 0.274 | 0.445 |
| | ATC-NE | 0.747 | 0.659 | **0.701** | — | **0.690** | 0.792 | **0.717** | 0.505 | 0.610 | **0.776** |
| | ATC-MC | **0.759** | 0.642 | 0.699 | — | 0.687 | 0.784 | 0.708 | 0.507 | 0.594 | 0.765 |
| movies | NI-SSMBA | 0.541 | 0.640 | 0.658 | 0.615 | — | 0.633 | 0.656 | 0.583 | 0.653 | 0.741 |
| | NI-EDA | 0.542 | 0.662 | 0.571 | **0.676** | — | 0.629 | 0.594 | 0.574 | 0.636 | 0.789 |
| | NI-BT | 0.602 | 0.679 | 0.677 | 0.653 | — | 0.723 | **0.739** | **0.700** | 0.709 | 0.754 |
| | NI-RandRep | 0.194 | 0.276 | 0.281 | 0.279 | — | 0.228 | 0.316 | 0.216 | 0.316 | 0.369 |
| | ATC-NE | 0.698 | 0.719 | 0.708 | 0.668 | — | 0.718 | 0.707 | 0.611 | 0.752 | 0.818 |
| | ATC-MC | **0.707** | **0.725** | **0.713** | 0.664 | — | **0.732** | 0.711 | 0.624 | **0.755** | **0.820** |
| pets | NI-SSMBA | 0.458 | 0.543 | 0.578 | 0.529 | 0.548 | — | 0.606 | 0.528 | 0.590 | 0.721 |
| | NI-EDA | 0.426 | 0.552 | 0.554 | 0.467 | 0.513 | — | 0.546 | 0.463 | 0.549 | 0.677 |
| | NI-BT | 0.485 | 0.523 | 0.477 | 0.549 | 0.563 | — | 0.581 | 0.550 | 0.580 | 0.587 |
| | NI-RandRep | 0.260 | 0.333 | 0.284 | 0.302 | 0.320 | — | 0.304 | 0.271 | 0.345 | 0.432 |
| | ATC-NE | 0.641 | **0.851** | **0.849** | **0.651** | 0.680 | — | **0.825** | 0.591 | **0.812** | **0.883** |
| | ATC-MC | **0.644** | 0.846 | 0.848 | 0.648 | **0.683** | — | 0.824 | 0.589 | 0.810 | **0.883** |
| sports | NI-SSMBA | 0.499 | 0.530 | 0.569 | 0.524 | 0.463 | 0.573 | — | 0.517 | 0.590 | 0.721 |
| | NI-EDA | 0.464 | 0.514 | 0.460 | 0.489 | 0.381 | 0.508 | — | 0.448 | 0.541 | 0.594 |
| | NI-BT | 0.406 | 0.295 | 0.334 | 0.383 | 0.381 | 0.325 | — | 0.331 | 0.393 | 0.411 |
| | NI-RandRep | 0.283 | 0.173 | 0.212 | 0.256 | 0.219 | 0.239 | — | 0.204 | 0.231 | 0.331 |
| | ATC-NE | 0.712 | **0.900** | **0.926** | 0.744 | 0.656 | 0.891 | — | 0.828 | 0.931 | **0.953** |
| | ATC-MC | **0.723** | 0.897 | **0.926** | 0.754 | **0.665** | **0.896** | — | **0.833** | **0.933** | 0.952 |
| tech | NI-SSMBA | 0.610 | 0.508 | 0.635 | 0.620 | 0.596 | 0.604 | 0.636 | — | 0.545 | 0.848 |
| | NI-EDA | 0.550 | 0.475 | 0.519 | 0.603 | 0.608 | 0.581 | 0.587 | — | 0.532 | 0.649 |
| | NI-BT | 0.588 | 0.447 | 0.572 | 0.632 | 0.597 | 0.558 | 0.603 | — | 0.523 | 0.648 |
| | NI-RandRep | 0.340 | 0.241 | 0.262 | 0.307 | 0.337 | 0.285 | 0.291 | — | 0.200 | 0.432 |
| | ATC-NE | 0.766 | 0.820 | **0.904** | 0.791 | 0.817 | **0.866** | 0.874 | — | **0.882** | **0.893** |
| | ATC-MC | **0.771** | **0.823** | **0.904** | **0.794** | 0.817 | 0.864 | **0.875** | — | **0.882** | **0.893** |
| tools | NI-SSMBA | 0.554 | 0.505 | 0.548 | 0.556 | 0.605 | 0.651 | 0.606 | 0.577 | — | 0.660 |
| | NI-EDA | 0.466 | 0.413 | 0.460 | 0.459 | 0.589 | 0.587 | 0.512 | 0.465 | — | 0.593 |
| | NI-BT | 0.451 | 0.385 | 0.447 | 0.476 | 0.482 | 0.501 | 0.483 | 0.484 | — | 0.462 |
| | NI-RandRep | 0.323 | 0.241 | 0.204 | 0.306 | 0.399 | 0.316 | 0.253 | 0.227 | — | 0.297 |
| | ATC-NE | 0.661 | **0.875** | 0.901 | 0.679 | 0.719 | **0.867** | 0.908 | 0.795 | — | **0.916** |
| | ATC-MC | **0.670** | 0.874 | **0.903** | 0.684 | **0.729** | 0.866 | **0.909** | 0.801 | — | **0.916** |
| toys | NI-SSMBA | 0.625 | 0.693 | 0.656 | 0.620 | 0.643 | 0.661 | 0.708 | **0.618** | 0.650 | — |
| | NI-EDA | 0.473 | 0.582 | 0.487 | 0.481 | 0.498 | 0.552 | 0.535 | 0.429 | 0.528 | — |
| | NI-BT | 0.311 | 0.408 | 0.331 | 0.351 | 0.324 | 0.355 | 0.420 | 0.334 | 0.357 | — |
| | NI-RandRep | 0.272 | 0.307 | 0.234 | 0.254 | 0.286 | 0.257 | 0.301 | 0.215 | 0.248 | — |
| | ATC-NE | 0.686 | **0.936** | 0.855 | 0.742 | 0.712 | 0.838 | 0.885 | 0.561 | 0.866 | — |
| | ATC-MC | **0.691** | 0.935 | **0.858** | 0.742 | **0.718** | **0.843** | **0.886** | 0.574 | **0.869** | — |

Table 12: Full $R^2$ metrics for all pairs of training and test domains for CNN models trained on AWS.

| Train Domain | Measure | Test Domain | | | | | | | | | |
|---|---|---|---|---|---|---|---|---|---|---|---|
| | | books | clothing | home | kindle | movies | pets | sports | tech | tools | toys |
| books | NI-SSMBA | — | **0.973** | **0.983** | **0.987** | 0.963 | **0.973** | **0.985** | **0.985** | 0.977 | 0.968 |
| | NI-EDA | — | 0.961 | 0.970 | 0.982 | **0.966** | 0.966 | 0.970 | 0.957 | **0.978** | **0.981** |
| | NI-BT | — | 0.946 | 0.979 | 0.985 | 0.961 | 0.972 | 0.972 | 0.975 | 0.970 | 0.955 |
| | NI-RandRep | — | 0.956 | 0.970 | 0.981 | 0.958 | 0.967 | 0.970 | 0.973 | 0.952 | 0.961 |
| | ATC-NE | — | 0.845 | 0.861 | 0.957 | 0.927 | 0.940 | 0.852 | 0.925 | 0.853 | 0.876 |
| | ATC-MC | — | 0.838 | 0.842 | 0.947 | 0.924 | 0.929 | 0.840 | 0.927 | 0.845 | 0.876 |
| clothing | NI-SSMBA | 0.944 | — | 0.974 | **0.942** | 0.980 | 0.981 | 0.984 | 0.973 | 0.969 | 0.978 |
| | NI-EDA | **0.974** | — | 0.970 | 0.922 | 0.955 | 0.978 | 0.961 | 0.971 | 0.953 | 0.972 |
| | NI-BT | 0.944 | — | **0.977** | 0.921 | 0.971 | **0.983** | 0.982 | 0.980 | 0.970 | **0.993** |
| | NI-RandRep | 0.950 | — | 0.965 | 0.933 | **0.982** | 0.977 | **0.985** | **0.981** | **0.974** | 0.973 |
| | ATC-NE | 0.792 | — | 0.934 | 0.764 | 0.913 | 0.939 | 0.933 | 0.846 | 0.914 | 0.973 |
| | ATC-MC | 0.801 | — | 0.945 | 0.789 | 0.917 | 0.948 | 0.933 | 0.887 | 0.926 | 0.971 |
| home | NI-SSMBA | 0.973 | 0.972 | — | **0.976** | **0.972** | **0.979** | 0.976 | **0.980** | 0.972 | **0.984** |
| | NI-EDA | 0.961 | 0.975 | — | 0.967 | 0.959 | 0.960 | 0.949 | 0.917 | 0.963 | 0.979 |
| | NI-BT | 0.947 | 0.958 | — | 0.944 | 0.947 | 0.948 | **0.977** | 0.972 | 0.971 | 0.974 |
| | NI-RandRep | **0.978** | **0.976** | — | 0.970 | 0.967 | 0.975 | 0.973 | 0.964 | **0.975** | 0.983 |
| | ATC-NE | 0.878 | 0.959 | — | 0.958 | 0.908 | 0.930 | 0.962 | 0.860 | 0.928 | 0.951 |
| | ATC-MC | 0.885 | 0.960 | — | 0.963 | 0.913 | 0.945 | 0.964 | 0.872 | 0.938 | 0.952 |
| kindle | NI-SSMBA | **0.992** | 0.963 | 0.966 | — | 0.961 | 0.957 | **0.963** | 0.971 | 0.943 | **0.984** |
| | NI-EDA | 0.983 | **0.968** | 0.951 | — | 0.966 | **0.962** | 0.953 | **0.974** | **0.963** | 0.969 |
| | NI-BT | 0.973 | 0.947 | **0.975** | — | 0.954 | 0.949 | 0.954 | 0.973 | 0.923 | 0.973 |
| | NI-RandRep | 0.979 | 0.946 | 0.956 | — | **0.967** | 0.961 | 0.946 | 0.970 | 0.934 | 0.975 |
| | ATC-NE | 0.931 | 0.380 | 0.510 | — | 0.861 | 0.162 | 0.260 | 0.424 | 0.179 | 0.608 |
| | ATC-MC | 0.930 | 0.534 | 0.623 | — | 0.871 | 0.294 | 0.392 | 0.551 | 0.316 | 0.672 |
| movies | NI-SSMBA | **0.984** | **0.976** | **0.969** | 0.981 | — | **0.962** | **0.983** | 0.958 | 0.961 | 0.971 |
| | NI-EDA | 0.974 | 0.975 | 0.947 | **0.985** | — | 0.928 | 0.977 | 0.950 | **0.972** | 0.969 |
| | NI-BT | 0.971 | 0.952 | 0.965 | 0.965 | — | 0.951 | 0.970 | 0.957 | 0.950 | **0.977** |
| | NI-RandRep | 0.981 | 0.969 | 0.954 | 0.971 | — | 0.958 | 0.969 | 0.957 | 0.958 | 0.962 |
| | ATC-NE | 0.948 | 0.864 | 0.902 | 0.949 | — | 0.746 | 0.908 | 0.813 | 0.817 | 0.917 |
| | ATC-MC | 0.949 | 0.893 | 0.928 | 0.948 | — | 0.802 | 0.929 | 0.861 | 0.859 | 0.926 |
| pets | NI-SSMBA | 0.943 | 0.974 | 0.981 | 0.945 | 0.942 | — | 0.979 | **0.982** | 0.978 | 0.969 |
| | NI-EDA | **0.965** | **0.980** | **0.985** | **0.958** | **0.952** | — | **0.983** | 0.975 | 0.976 | **0.983** |
| | NI-BT | 0.936 | 0.971 | 0.974 | 0.931 | 0.926 | — | 0.980 | 0.963 | **0.979** | 0.977 |
| | NI-RandRep | 0.941 | 0.976 | 0.971 | 0.925 | 0.931 | — | 0.976 | 0.968 | 0.976 | 0.979 |
| | ATC-NE | 0.594 | 0.967 | 0.900 | 0.577 | 0.862 | — | 0.947 | 0.959 | 0.929 | 0.882 |
| | ATC-MC | 0.611 | 0.968 | 0.909 | 0.589 | 0.851 | — | 0.955 | 0.959 | 0.939 | 0.894 |
| sports | NI-SSMBA | 0.979 | 0.982 | **0.994** | **0.980** | 0.979 | 0.970 | — | 0.982 | 0.984 | 0.978 |
| | NI-EDA | 0.979 | **0.987** | 0.986 | 0.960 | **0.984** | **0.985** | — | 0.985 | **0.986** | 0.987 |
| | NI-BT | 0.966 | 0.977 | 0.989 | 0.949 | 0.969 | 0.979 | — | **0.986** | 0.984 | **0.988** |
| | NI-RandRep | **0.981** | 0.979 | **0.994** | 0.978 | 0.980 | 0.978 | — | 0.985 | 0.985 | 0.984 |
| | ATC-NE | 0.655 | 0.945 | 0.963 | 0.688 | 0.850 | 0.890 | — | 0.938 | 0.969 | 0.957 |
| | ATC-MC | 0.705 | 0.951 | 0.965 | 0.722 | 0.882 | 0.905 | — | 0.944 | 0.972 | 0.953 |
| tech | NI-SSMBA | **0.944** | 0.979 | 0.977 | **0.947** | **0.967** | 0.958 | 0.975 | — | **0.990** | 0.981 |
| | NI-EDA | 0.917 | 0.969 | 0.971 | 0.925 | 0.953 | 0.937 | 0.967 | — | 0.963 | 0.972 |
| | NI-BT | 0.894 | **0.982** | **0.987** | 0.924 | 0.937 | 0.933 | **0.978** | — | 0.987 | 0.972 |
| | NI-RandRep | 0.929 | 0.959 | 0.977 | 0.918 | 0.952 | 0.944 | 0.963 | — | 0.984 | 0.971 |
| | ATC-NE | 0.937 | 0.939 | 0.983 | 0.933 | 0.946 | **0.961** | 0.961 | — | 0.974 | 0.981 |
| | ATC-MC | 0.941 | 0.941 | 0.980 | 0.928 | 0.955 | 0.958 | 0.963 | — | 0.971 | **0.982** |
| tools | NI-SSMBA | **0.977** | **0.988** | 0.975 | 0.967 | 0.971 | 0.986 | 0.985 | **0.978** | — | 0.983 |
| | NI-EDA | 0.975 | 0.985 | **0.978** | **0.969** | **0.980** | 0.986 | **0.988** | 0.975 | — | 0.982 |
| | NI-BT | 0.940 | 0.976 | 0.970 | 0.941 | 0.944 | 0.969 | 0.978 | 0.948 | — | 0.975 |
| | NI-RandRep | 0.963 | 0.985 | **0.978** | 0.954 | 0.963 | **0.988** | 0.984 | 0.977 | — | **0.987** |
| | ATC-NE | 0.857 | 0.945 | 0.964 | 0.883 | 0.865 | 0.925 | 0.964 | 0.963 | — | 0.922 |
| | ATC-MC | 0.868 | 0.941 | 0.965 | 0.887 | 0.876 | 0.936 | 0.967 | 0.971 | — | 0.919 |
| toys | NI-SSMBA | 0.955 | **0.971** | **0.985** | **0.980** | 0.961 | 0.966 | **0.988** | **0.978** | 0.971 | — |
| | NI-EDA | **0.965** | 0.968 | 0.984 | 0.970 | **0.985** | **0.973** | 0.981 | 0.975 | **0.983** | — |
| | NI-BT | 0.890 | 0.946 | 0.959 | 0.944 | 0.925 | 0.935 | 0.965 | 0.943 | 0.934 | — |
| | NI-RandRep | 0.959 | 0.970 | 0.979 | 0.960 | 0.965 | 0.967 | 0.977 | **0.978** | 0.974 | — |
| | ATC-NE | 0.883 | 0.878 | 0.885 | 0.763 | 0.906 | 0.798 | 0.938 | 0.815 | 0.899 | — |
| | ATC-MC | 0.889 | 0.897 | 0.895 | 0.754 | 0.920 | 0.818 | 0.947 | 0.833 | 0.914 | — |

Table 13: Full $R^2$ metrics for all pairs of training and test domains for BERT models trained on AWS.

| Train Domain | Measure | slate | verbatim | facetoface | oup | nineeleven | fiction | telephone | travel | letters | government |
|---|---|---|---|---|---|---|---|---|---|---|---|
| | | | | | | Test Domain | | | | | |
| slate | NI-SSMBA | — | 0.457 | 0.760 | 0.544 | 0.642 | 0.706 | 0.727 | 0.585 | 0.640 | 0.714 |
| | NI-EDA | — | 0.574 | 0.723 | 0.565 | 0.620 | 0.708 | 0.731 | 0.549 | 0.595 | 0.706 |
| | NI-BT | — | **0.862** | **0.920** | **0.942** | **0.930** | **0.913** | **0.878** | **0.936** | **0.949** | **0.940** |
| | NI-RandRep | — | 0.242 | 0.336 | 0.501 | 0.415 | 0.121 | 0.170 | 0.301 | 0.652 | 0.470 |
| | ATC-NE | — | 0.681 | 0.677 | 0.594 | 0.599 | 0.741 | 0.669 | 0.664 | 0.661 | 0.773 |
| | ATC-MC | — | 0.682 | 0.675 | 0.590 | 0.600 | 0.737 | 0.670 | 0.665 | 0.661 | 0.771 |
| fiction | NI-SSMBA | 0.581 | 0.492 | 0.765 | 0.451 | 0.530 | — | 0.683 | 0.499 | 0.502 | 0.614 |
| | NI-EDA | 0.657 | 0.601 | 0.728 | 0.620 | 0.530 | — | 0.676 | 0.527 | 0.508 | 0.642 |
| | NI-BT | **0.889** | **0.887** | **0.917** | **0.950** | **0.921** | — | **0.863** | **0.928** | **0.935** | **0.941** |
| | NI-RandRep | 0.360 | 0.414 | 0.459 | 0.666 | 0.531 | — | 0.188 | 0.464 | 0.680 | 0.631 |
| | ATC-NE | 0.530 | 0.425 | 0.718 | 0.323 | 0.536 | — | 0.556 | 0.425 | 0.597 | 0.522 |
| | ATC-MC | 0.525 | 0.418 | 0.719 | 0.326 | 0.534 | — | 0.555 | 0.422 | 0.592 | 0.521 |
| telephone | NI-SSMBA | 0.539 | 0.453 | 0.826 | 0.429 | 0.480 | 0.647 | — | 0.436 | 0.559 | 0.439 |
| | NI-EDA | 0.677 | 0.636 | 0.804 | 0.492 | 0.639 | 0.819 | — | 0.557 | 0.653 | 0.568 |
| | NI-BT | **0.882** | **0.901** | **0.928** | **0.936** | **0.927** | **0.912** | — | **0.937** | **0.916** | **0.919** |
| | NI-RandRep | 0.332 | 0.498 | 0.488 | 0.584 | 0.538 | 0.310 | — | 0.586 | 0.677 | 0.576 |
| | ATC-NE | 0.557 | 0.518 | 0.762 | 0.527 | 0.467 | 0.695 | — | 0.368 | 0.554 | 0.468 |
| | ATC-MC | 0.561 | 0.518 | 0.762 | 0.524 | 0.467 | 0.698 | — | 0.372 | 0.560 | 0.470 |
| travel | NI-SSMBA | 0.503 | 0.506 | 0.588 | 0.419 | 0.501 | 0.543 | 0.597 | — | 0.670 | 0.507 |
| | NI-EDA | 0.444 | 0.429 | 0.502 | 0.388 | 0.342 | 0.532 | 0.478 | — | 0.438 | 0.377 |
| | NI-BT | **0.806** | **0.799** | **0.861** | **0.916** | **0.889** | **0.828** | **0.802** | — | **0.923** | **0.907** |
| | NI-RandRep | 0.315 | 0.333 | 0.385 | 0.643 | 0.516 | 0.182 | 0.224 | — | 0.690 | 0.626 |
| | ATC-NE | 0.561 | 0.516 | 0.498 | 0.686 | 0.546 | 0.581 | 0.606 | — | 0.585 | 0.598 |
| | ATC-MC | 0.563 | 0.517 | 0.503 | 0.690 | 0.544 | 0.588 | 0.611 | — | 0.582 | 0.607 |
| government | NI-SSMBA | 0.592 | 0.497 | 0.618 | 0.572 | 0.596 | 0.648 | 0.593 | 0.539 | 0.676 | — |
| | NI-EDA | 0.577 | 0.574 | 0.550 | 0.476 | 0.618 | 0.600 | 0.491 | 0.518 | 0.526 | — |
| | NI-BT | **0.818** | **0.812** | **0.859** | **0.909** | **0.893** | **0.813** | **0.809** | **0.909** | **0.917** | — |
| | NI-RandRep | 0.010 | 0.083 | 0.207 | 0.306 | 0.279 | 0.011 | 0.027 | 0.123 | 0.360 | — |
| | ATC-NE | 0.663 | 0.405 | 0.564 | 0.722 | 0.337 | 0.634 | 0.630 | 0.509 | 0.653 | — |
| | ATC-MC | 0.666 | 0.405 | 0.574 | 0.721 | 0.341 | 0.633 | 0.636 | 0.509 | 0.654 | — |

Table 14: Full $R^2$ metrics for all pairs of training and test domains for CNN models trained on MNLI.

| Train Domain | Measure | Test Domain | | | | | | | | | |
|---|---|---|---|---|---|---|---|---|---|---|---|
| | | slate | verbatim | facetoface | oup | nineeleven | fiction | telephone | travel | letters | government |
| slate | NI-SSMBA | — | **0.920** | 0.905 | 0.943 | 0.916 | 0.929 | **0.930** | **0.957** | 0.931 | **0.970** |
| | NI-EDA | — | 0.701 | 0.754 | 0.816 | 0.784 | 0.790 | 0.708 | 0.787 | 0.820 | 0.837 |
| | NI-BT | — | 0.862 | **0.920** | 0.942 | **0.930** | 0.913 | 0.878 | 0.936 | **0.949** | 0.940 |
| | NI-RandRep | — | 0.242 | 0.336 | 0.501 | 0.415 | 0.121 | 0.170 | 0.301 | 0.652 | 0.470 |
| | ATC-NE | — | 0.878 | 0.899 | **0.954** | 0.911 | 0.925 | 0.911 | 0.904 | 0.890 | 0.951 |
| | ATC-MC | — | 0.871 | 0.884 | 0.953 | 0.908 | **0.932** | 0.914 | 0.911 | 0.887 | 0.949 |
| fiction | NI-SSMBA | 0.954 | 0.928 | 0.895 | 0.952 | 0.946 | — | 0.952 | 0.975 | 0.946 | 0.954 |
| | NI-EDA | 0.661 | 0.687 | 0.699 | 0.840 | 0.686 | — | 0.683 | 0.764 | 0.799 | 0.797 |
| | NI-BT | 0.889 | 0.887 | **0.917** | 0.950 | 0.921 | — | 0.863 | 0.928 | 0.935 | 0.941 |
| | NI-RandRep | 0.360 | 0.414 | 0.459 | 0.666 | 0.531 | — | 0.188 | 0.464 | 0.680 | 0.631 |
| | ATC-NE | 0.808 | 0.585 | 0.881 | 0.726 | 0.939 | — | 0.836 | 0.826 | 0.850 | 0.890 |
| | ATC-MC | 0.817 | 0.652 | 0.889 | 0.732 | 0.936 | — | 0.841 | 0.830 | 0.861 | 0.892 |
| telephone | NI-SSMBA | **0.943** | **0.962** | **0.948** | **0.953** | 0.917 | **0.954** | — | 0.921 | **0.934** | **0.956** |
| | NI-EDA | 0.776 | 0.787 | 0.790 | 0.852 | 0.823 | 0.847 | — | 0.853 | 0.877 | 0.852 |
| | NI-BT | 0.882 | 0.901 | 0.928 | 0.936 | **0.927** | 0.912 | — | **0.937** | 0.916 | 0.919 |
| | NI-RandRep | 0.332 | 0.498 | 0.488 | 0.584 | 0.538 | 0.310 | — | 0.586 | 0.677 | 0.576 |
| | ATC-NE | 0.662 | 0.576 | 0.884 | 0.904 | 0.762 | 0.665 | — | 0.771 | 0.856 | 0.889 |
| | ATC-MC | 0.714 | 0.668 | 0.902 | 0.924 | 0.807 | 0.732 | — | 0.795 | 0.857 | 0.899 |
| travel | NI-SSMBA | **0.938** | **0.935** | **0.949** | **0.924** | **0.948** | **0.946** | **0.958** | — | **0.962** | **0.953** |
| | NI-EDA | 0.538 | 0.553 | 0.471 | 0.711 | 0.602 | 0.638 | 0.533 | — | 0.755 | 0.699 |
| | NI-BT | 0.806 | 0.799 | 0.861 | 0.916 | 0.889 | 0.828 | 0.802 | — | 0.923 | 0.907 |
| | NI-RandRep | 0.315 | 0.333 | 0.385 | 0.643 | 0.516 | 0.182 | 0.224 | — | 0.690 | 0.626 |
| | ATC-NE | 0.319 | 0.417 | 0.500 | 0.889 | 0.768 | 0.396 | 0.494 | — | 0.928 | 0.911 |
| | ATC-MC | 0.412 | 0.506 | 0.591 | 0.906 | 0.789 | 0.525 | 0.572 | — | 0.921 | 0.917 |
| government | NI-SSMBA | **0.916** | **0.960** | 0.781 | **0.919** | 0.816 | **0.882** | **0.914** | **0.950** | **0.920** | — |
| | NI-EDA | 0.484 | 0.572 | 0.544 | 0.728 | 0.715 | 0.655 | 0.566 | 0.644 | 0.681 | — |
| | NI-BT | 0.818 | 0.812 | **0.859** | 0.909 | **0.893** | 0.813 | 0.809 | 0.909 | 0.917 | — |
| | NI-RandRep | 0.010 | 0.083 | 0.207 | 0.306 | 0.279 | 0.011 | 0.027 | 0.123 | 0.360 | — |
| | ATC-NE | 0.544 | 0.385 | 0.439 | 0.914 | 0.731 | 0.157 | 0.449 | 0.538 | 0.842 | — |
| | ATC-MC | 0.624 | 0.481 | 0.478 | **0.919** | 0.797 | 0.243 | 0.505 | 0.576 | 0.856 | — |

Table 15: Full $R^2$ metrics for all pairs of training and test domains for BERT models trained on MNLI.

| Measure | Test Domain | | | | | | |
|---|---|---|---|---|---|---|---|
| | ImageNetV2 | ImagenNet-Sketch | ObjectNet | ImageNet-Vid | YTBB | ImageNet-R | ImageNet-A |
| NI-RandAug | 0.816 | 0.677 | 0.646 | **0.790** | **0.692** | **0.724** | 0.586 |
| NI-Translate | 0.785 | 0.515 | 0.516 | 0.685 | 0.544 | 0.581 | 0.439 |
| NI-Erase | 0.779 | 0.322 | 0.52 | 0.706 | 0.560 | 0.444 | 0.517 |
| NI-FC | 0.842 | 0.552 | 0.645 | 0.780 | 0.649 | 0.679 | **0.589** |
| ATC-MC | 0.891 | 0.708 | **0.756** | 0.761 | 0.509 | 0.521 | 0.190 |
| ATC-NE | **0.926** | **0.717** | 0.671 | 0.651 | 0.494 | 0.569 | 0.248 |

Table 16: Full macro $\tau$ metrics for all ImageNet test domains. We average values across models.

| Model | Train Domain | Measure | Test Domain | | |
|---|---|---|---|---|---|
| | | | SVHN | Colored MNIST | MNIST |
| NiN | SVHN | NI-RandAug | — | 0.668 | 0.616 |
| | | NI-Translate | — | **0.699** | 0.635 |
| | | NI-Erase | — | -0.102 | -0.168 |
| | | NI-FC | — | -0.233 | -0.398 |
| | | ATC-NE | — | 0.577 | 0.695 |
| | | ATC-MC | — | 0.646 | **0.716** |

Table 17: Full macro $\tau$ metrics for all test domains for image classification models on SVHN, Colored MNIST, and MNIST.

| Model | Train Domain | Measure | Test Domain | | | |
|---|---|---|---|---|---|---|
| | | | CIFAR10 | CINIC10 | CIFAR10.1 | CIFAR10.2 |
| NiN | CIFAR10 | NI-RandAug | — | **0.785** | **0.830** | **0.783** |
| | | NI-Translate | — | 0.621 | 0.739 | 0.644 |
| | | NI-Erase | — | 0.385 | 0.485 | 0.499 |
| | | NI-FC | — | 0.194 | 0.077 | -0.037 |
| | | ATC-NE | — | 0.250 | 0.543 | 0.484 |
| | | ATC-MC | — | 0.319 | 0.514 | 0.446 |
| ResNet | CIFAR10 | NI-RandAug | — | 0.706 | 0.722 | **0.742** |
| | | NI-Translate | — | **0.800** | **0.796** | 0.741 |
| | | NI-Erase | — | 0.723 | 0.727 | 0.699 |
| | | NI-FC | — | 0.647 | 0.585 | 0.607 |
| | | ATC-NE | — | 0.644 | 0.687 | 0.630 |
| | | ATC-MC | — | 0.631 | 0.672 | 0.624 |
| VGG | CIFAR10 | NI-RandAug | — | **0.868** | **0.831** | **0.772** |
| | | NI-Translate | — | 0.700 | 0.679 | 0.632 |
| | | NI-Erase | — | -0.153 | -0.196 | -0.215 |
| | | NI-FC | — | 0.597 | 0.569 | 0.512 |
| | | ATC-NE | — | 0.531 | 0.656 | 0.599 |
| | | ATC-MC | — | 0.533 | 0.648 | 0.586 |
| CNN | CINIC10 | NI-RandAug | **0.756** | — | **0.698** | **0.695** |
| | | NI-Translate | 0.347 | — | 0.318 | 0.267 |
| | | NI-Erase | 0.619 | — | 0.539 | 0.577 |
| | | NI-FC | 0.249 | — | 0.286 | 0.252 |
| | | ATC-NE | 0.607 | — | 0.510 | 0.447 |
| | | ATC-MC | 0.611 | — | 0.500 | 0.440 |

Table 18: Full macro $\tau$ metrics for all test domains for image classification models on CIFAR10, CINIC10, CIFAR10.1, and CIFAR10.2.

| Train Domain | Measure | Test Domain | | | | | | | | | |
|---|---|---|---|---|---|---|---|---|---|---|---|
| | | books | clothing | home | kindle | movies | pets | sports | tech | tools | toys |
| books | NI-SSMBA | — | 0.708 | **0.741** | **0.687** | 0.678 | 0.754 | **0.738** | **0.738** | **0.721** | 0.669 |
| | NI-EDA | — | **0.722** | 0.738 | 0.640 | **0.685** | **0.756** | 0.688 | 0.709 | 0.719 | **0.725** |
| | NI-BT | — | 0.615 | 0.631 | 0.549 | 0.596 | 0.656 | 0.602 | 0.644 | 0.584 | 0.621 |
| | NI-RandRep | — | 0.591 | 0.586 | 0.499 | 0.587 | 0.587 | 0.530 | 0.530 | 0.546 | 0.589 |
| | ATC-NE | — | 0.605 | 0.642 | 0.534 | 0.496 | 0.629 | 0.605 | 0.538 | 0.618 | 0.673 |
| | ATC-MC | — | 0.604 | 0.639 | 0.534 | 0.489 | 0.630 | 0.606 | 0.544 | 0.616 | 0.670 |
| clothing | NI-SSMBA | 0.769 | — | **0.774** | 0.773 | **0.793** | 0.726 | 0.653 | **0.716** | **0.692** | 0.709 |
| | NI-EDA | 0.681 | — | 0.670 | **0.787** | 0.721 | 0.696 | **0.675** | 0.694 | 0.690 | **0.784** |
| | NI-BT | 0.574 | — | 0.542 | 0.654 | 0.595 | 0.481 | 0.506 | 0.501 | 0.503 | 0.679 |
| | NI-RandRep | 0.623 | — | 0.591 | 0.603 | 0.612 | 0.557 | 0.496 | 0.507 | 0.549 | 0.640 |
| | ATC-NE | 0.420 | — | 0.366 | 0.408 | 0.450 | 0.308 | 0.501 | 0.242 | 0.560 | 0.563 |
| | ATC-MC | 0.421 | — | 0.363 | 0.409 | 0.451 | 0.315 | 0.498 | 0.239 | 0.558 | 0.563 |
| home | NI-SSMBA | 0.786 | 0.552 | — | 0.731 | **0.731** | 0.769 | **0.790** | **0.789** | **0.775** | 0.614 |
| | NI-EDA | 0.734 | **0.659** | — | **0.767** | 0.649 | 0.757 | 0.707 | 0.723 | 0.696 | **0.748** |
| | NI-BT | 0.670 | 0.498 | — | 0.641 | 0.639 | 0.634 | 0.706 | 0.688 | 0.662 | 0.630 |
| | NI-RandRep | 0.700 | 0.549 | — | 0.665 | 0.705 | 0.686 | 0.663 | 0.664 | 0.632 | 0.651 |
| | ATC-NE | 0.450 | 0.480 | — | 0.459 | 0.447 | 0.455 | 0.559 | 0.497 | 0.558 | 0.445 |
| | ATC-MC | 0.459 | 0.480 | — | 0.460 | 0.452 | 0.458 | 0.557 | 0.495 | 0.557 | 0.448 |
| kindle | NI-SSMBA | 0.546 | **0.679** | 0.648 | — | 0.557 | 0.699 | 0.656 | 0.608 | 0.607 | 0.630 |
| | NI-EDA | **0.663** | 0.644 | **0.717** | — | **0.637** | **0.743** | **0.699** | **0.676** | **0.643** | **0.696** |
| | NI-BT | 0.556 | 0.648 | 0.652 | — | 0.562 | 0.686 | 0.683 | 0.668 | **0.643** | 0.674 |
| | NI-RandRep | 0.454 | 0.519 | 0.406 | — | 0.428 | 0.555 | 0.434 | 0.415 | 0.415 | 0.535 |
| | ATC-NE | 0.370 | 0.517 | 0.468 | — | 0.338 | 0.583 | 0.540 | 0.383 | 0.472 | 0.545 |
| | ATC-MC | 0.388 | 0.512 | 0.469 | — | 0.342 | 0.578 | 0.538 | 0.384 | 0.467 | 0.541 |
| movies | NI-SSMBA | 0.572 | 0.689 | **0.717** | **0.660** | — | 0.732 | **0.741** | 0.675 | 0.714 | **0.749** |
| | NI-EDA | 0.500 | **0.717** | 0.711 | 0.629 | — | 0.733 | 0.725 | 0.660 | **0.719** | 0.748 |
| | NI-BT | **0.619** | 0.681 | 0.687 | 0.595 | — | **0.738** | 0.729 | **0.717** | 0.717 | 0.717 |
| | NI-RandRep | 0.435 | 0.491 | 0.523 | 0.554 | — | 0.460 | 0.528 | 0.436 | 0.517 | 0.570 |
| | ATC-NE | 0.436 | 0.605 | 0.623 | 0.398 | — | 0.571 | 0.622 | 0.502 | 0.659 | 0.701 |
| | ATC-MC | 0.449 | 0.612 | 0.631 | 0.403 | — | 0.586 | 0.628 | 0.520 | 0.661 | 0.704 |
| pets | NI-SSMBA | 0.747 | 0.560 | 0.653 | 0.766 | 0.769 | — | **0.719** | **0.696** | 0.697 | 0.731 |
| | NI-EDA | **0.762** | **0.684** | **0.682** | **0.794** | 0.773 | — | 0.709 | 0.666 | 0.689 | **0.745** |
| | NI-BT | 0.741 | 0.550 | 0.625 | 0.751 | **0.776** | — | 0.704 | 0.676 | **0.702** | 0.654 |
| | NI-RandRep | 0.619 | 0.575 | 0.572 | 0.696 | 0.662 | — | 0.580 | 0.532 | 0.590 | 0.710 |
| | ATC-NE | 0.570 | 0.545 | 0.462 | 0.533 | 0.543 | — | 0.511 | 0.333 | 0.503 | 0.634 |
| | ATC-MC | 0.575 | 0.541 | 0.461 | 0.535 | 0.546 | — | 0.510 | 0.332 | 0.502 | 0.632 |
| sports | NI-SSMBA | **0.774** | 0.561 | 0.657 | **0.812** | **0.764** | 0.683 | — | 0.689 | **0.746** | 0.651 |
| | NI-EDA | 0.770 | **0.633** | **0.659** | 0.791 | 0.710 | **0.712** | — | **0.701** | 0.732 | 0.692 |
| | NI-BT | 0.670 | 0.484 | 0.598 | 0.669 | 0.672 | 0.545 | — | 0.534 | 0.704 | 0.538 |
| | NI-RandRep | 0.684 | 0.472 | 0.562 | 0.704 | 0.638 | 0.589 | — | 0.614 | 0.622 | **0.693** |
| | ATC-NE | 0.502 | 0.456 | 0.454 | 0.506 | 0.443 | 0.374 | — | 0.397 | 0.565 | 0.577 |
| | ATC-MC | 0.513 | 0.459 | 0.459 | 0.521 | 0.453 | 0.393 | — | 0.410 | 0.574 | 0.574 |
| tech | NI-SSMBA | **0.784** | 0.674 | **0.768** | **0.793** | **0.779** | 0.755 | **0.785** | — | 0.690 | 0.746 |
| | NI-EDA | 0.718 | **0.698** | 0.760 | 0.743 | 0.752 | **0.817** | 0.772 | — | **0.726** | **0.781** |
| | NI-BT | 0.715 | 0.576 | 0.719 | 0.732 | 0.746 | 0.716 | 0.726 | — | 0.687 | 0.683 |
| | NI-RandRep | 0.680 | 0.647 | 0.652 | 0.686 | 0.668 | 0.677 | 0.664 | — | 0.553 | 0.742 |
| | ATC-NE | 0.547 | 0.688 | 0.681 | 0.560 | 0.640 | 0.653 | 0.666 | — | 0.637 | 0.723 |
| | ATC-MC | 0.553 | 0.690 | 0.678 | 0.567 | 0.639 | 0.650 | 0.664 | — | 0.633 | 0.721 |
| tools | NI-SSMBA | **0.783** | 0.541 | 0.604 | **0.777** | **0.782** | 0.757 | **0.758** | **0.730** | — | 0.682 |
| | NI-EDA | 0.716 | 0.544 | **0.628** | 0.769 | 0.735 | **0.764** | 0.722 | 0.696 | — | **0.745** |
| | NI-BT | 0.691 | 0.342 | 0.454 | 0.681 | 0.677 | 0.567 | 0.575 | 0.636 | — | 0.513 |
| | NI-RandRep | 0.661 | **0.550** | 0.581 | 0.675 | 0.685 | 0.668 | 0.623 | 0.584 | — | 0.687 |
| | ATC-NE | 0.621 | 0.472 | 0.480 | 0.647 | 0.648 | 0.536 | 0.598 | 0.448 | — | 0.619 |
| | ATC-MC | 0.630 | 0.465 | 0.491 | 0.649 | 0.655 | 0.538 | 0.597 | 0.456 | — | 0.622 |
| toys | NI-SSMBA | **0.760** | **0.702** | **0.758** | 0.763 | 0.731 | 0.775 | **0.782** | 0.750 | **0.753** | — |
| | NI-EDA | 0.689 | 0.678 | 0.661 | 0.757 | 0.646 | 0.736 | 0.711 | 0.661 | 0.700 | — |
| | NI-BT | 0.582 | 0.604 | 0.683 | 0.626 | 0.610 | 0.695 | 0.723 | 0.660 | 0.658 | — |
| | NI-RandRep | 0.629 | 0.660 | 0.575 | 0.661 | 0.609 | 0.605 | 0.599 | 0.543 | 0.582 | — |
| | ATC-NE | 0.401 | 0.617 | 0.309 | 0.479 | 0.370 | 0.323 | 0.485 | 0.143 | 0.526 | — |
| | ATC-MC | 0.408 | 0.614 | 0.315 | 0.476 | 0.368 | 0.329 | 0.488 | 0.146 | 0.535 | — |

Table 19: Full macro $\tau$ metrics for all pairs of training and test domains for CNN models trained on AWS.

| Train Domain | Measure | Test Domain | | | | | | | | | |
|---|---|---|---|---|---|---|---|---|---|---|---|
| | | books | clothing | home | kindle | movies | pets | sports | tech | tools | toys |
| books | NI-SSMBA | — | **0.886** | **0.906** | 0.893 | 0.901 | **0.900** | **0.925** | **0.918** | **0.931** | **0.857** |
| | NI-EDA | — | 0.883 | 0.851 | 0.890 | 0.852 | 0.863 | 0.854 | 0.855 | 0.895 | 0.844 |
| | NI-BT | — | 0.830 | 0.887 | 0.893 | 0.883 | 0.884 | 0.871 | 0.878 | 0.869 | 0.824 |
| | NI-RandRep | — | 0.882 | 0.871 | **0.908** | **0.906** | 0.897 | 0.894 | 0.900 | 0.913 | 0.846 |
| | ATC-NE | — | 0.803 | 0.814 | 0.851 | 0.821 | 0.885 | 0.756 | 0.843 | 0.774 | 0.832 |
| | ATC-MC | — | 0.789 | 0.806 | 0.850 | 0.794 | 0.874 | 0.760 | 0.845 | 0.768 | 0.824 |
| clothing | NI-SSMBA | 0.797 | — | 0.882 | 0.714 | 0.773 | 0.826 | **0.881** | 0.826 | **0.883** | **0.824** |
| | NI-EDA | **0.828** | — | 0.848 | **0.813** | 0.787 | **0.862** | 0.741 | **0.839** | 0.848 | 0.777 |
| | NI-BT | 0.817 | — | **0.903** | 0.686 | **0.837** | 0.832 | 0.849 | 0.835 | 0.823 | 0.778 |
| | NI-RandRep | 0.762 | — | 0.858 | 0.691 | 0.826 | 0.832 | 0.872 | 0.822 | 0.868 | 0.786 |
| | ATC-NE | 0.729 | — | 0.805 | 0.672 | 0.681 | 0.817 | 0.728 | 0.698 | 0.792 | 0.791 |
| | ATC-MC | 0.735 | — | 0.794 | 0.683 | 0.701 | 0.858 | 0.744 | 0.736 | 0.808 | 0.791 |
| home | NI-SSMBA | 0.736 | 0.800 | — | **0.786** | 0.780 | 0.868 | 0.896 | **0.876** | 0.888 | 0.885 |
| | NI-EDA | 0.680 | **0.885** | — | 0.768 | **0.830** | **0.880** | 0.879 | 0.843 | 0.877 | **0.930** |
| | NI-BT | **0.749** | 0.788 | — | 0.756 | 0.778 | 0.819 | 0.871 | 0.835 | **0.906** | 0.866 |
| | NI-RandRep | 0.729 | 0.811 | — | 0.701 | 0.759 | 0.853 | **0.931** | 0.831 | 0.876 | 0.874 |
| | ATC-NE | 0.691 | 0.788 | — | 0.750 | 0.802 | 0.791 | 0.797 | 0.739 | 0.824 | 0.824 |
| | ATC-MC | 0.684 | 0.782 | — | 0.738 | 0.793 | 0.806 | 0.772 | 0.751 | 0.826 | 0.819 |
| kindle | NI-SSMBA | 0.735 | **0.850** | 0.885 | — | 0.701 | **0.913** | 0.875 | 0.869 | 0.763 | **0.871** |
| | NI-EDA | **0.840** | 0.830 | 0.824 | — | 0.709 | 0.879 | 0.856 | 0.865 | 0.854 | 0.790 |
| | NI-BT | 0.727 | 0.848 | **0.892** | — | **0.763** | 0.856 | **0.926** | **0.871** | **0.879** | 0.861 |
| | NI-RandRep | 0.773 | 0.848 | 0.816 | — | 0.735 | 0.875 | 0.857 | 0.843 | 0.794 | 0.847 |
| | ATC-NE | 0.695 | 0.335 | 0.486 | — | 0.644 | 0.211 | 0.274 | 0.426 | 0.238 | 0.589 |
| | ATC-MC | 0.729 | 0.455 | 0.558 | — | 0.671 | 0.337 | 0.366 | 0.509 | 0.369 | 0.652 |
| movies | NI-SSMBA | 0.845 | **0.897** | **0.888** | 0.782 | — | 0.850 | **0.931** | 0.877 | 0.881 | 0.896 |
| | NI-EDA | 0.764 | 0.881 | 0.852 | **0.863** | — | **0.859** | 0.901 | 0.863 | **0.882** | **0.938** |
| | NI-BT | 0.810 | 0.894 | 0.882 | 0.719 | — | 0.837 | 0.894 | 0.864 | 0.872 | 0.870 |
| | NI-RandRep | **0.851** | 0.875 | 0.855 | 0.742 | — | 0.850 | 0.905 | 0.864 | 0.861 | 0.874 |
| | ATC-NE | 0.692 | 0.745 | 0.807 | 0.712 | — | 0.561 | 0.768 | 0.580 | 0.620 | 0.763 |
| | ATC-MC | 0.698 | 0.770 | 0.847 | 0.722 | — | 0.622 | 0.813 | 0.637 | 0.678 | 0.784 |
| pets | NI-SSMBA | 0.728 | 0.737 | 0.866 | 0.737 | 0.829 | — | **0.883** | 0.867 | 0.859 | 0.773 |
| | NI-EDA | 0.732 | **0.831** | **0.880** | 0.601 | 0.788 | — | 0.853 | 0.834 | **0.903** | **0.841** |
| | NI-BT | 0.697 | 0.685 | 0.816 | **0.758** | 0.816 | — | 0.867 | 0.814 | 0.865 | 0.810 |
| | NI-RandRep | **0.753** | 0.731 | 0.813 | 0.735 | **0.841** | — | 0.828 | **0.871** | 0.869 | 0.774 |
| | ATC-NE | 0.623 | 0.740 | 0.719 | 0.553 | 0.737 | — | 0.815 | 0.788 | 0.748 | 0.781 |
| | ATC-MC | 0.662 | 0.753 | 0.723 | 0.566 | 0.711 | — | 0.831 | 0.781 | 0.752 | 0.750 |
| sports | NI-SSMBA | 0.715 | 0.743 | **0.879** | **0.764** | **0.837** | 0.863 | — | **0.879** | **0.860** | **0.870** |
| | NI-EDA | **0.757** | 0.731 | 0.807 | 0.673 | 0.798 | 0.786 | — | 0.815 | 0.827 | 0.833 |
| | NI-BT | 0.728 | **0.753** | 0.766 | 0.664 | 0.826 | 0.832 | — | 0.783 | 0.824 | 0.865 |
| | NI-RandRep | 0.735 | 0.702 | 0.816 | 0.728 | 0.808 | 0.802 | — | 0.830 | 0.849 | 0.836 |
| | ATC-NE | 0.489 | 0.734 | 0.738 | 0.567 | 0.720 | 0.726 | — | 0.726 | 0.725 | 0.862 |
| | ATC-MC | 0.484 | 0.728 | 0.766 | 0.562 | 0.754 | 0.727 | — | 0.725 | 0.741 | 0.860 |
| tech | NI-SSMBA | 0.784 | 0.763 | 0.845 | 0.710 | 0.769 | 0.853 | 0.755 | — | 0.886 | **0.877** |
| | NI-EDA | 0.771 | **0.867** | **0.881** | **0.763** | 0.801 | 0.855 | **0.886** | — | **0.904** | 0.869 |
| | NI-BT | 0.753 | 0.750 | 0.867 | 0.719 | 0.805 | 0.869 | 0.787 | — | 0.842 | 0.835 |
| | NI-RandRep | **0.819** | 0.721 | 0.843 | 0.711 | **0.854** | **0.889** | 0.787 | — | 0.854 | 0.874 |
| | ATC-NE | 0.737 | 0.744 | 0.787 | 0.751 | 0.732 | 0.795 | 0.727 | — | 0.830 | 0.863 |
| | ATC-MC | 0.771 | 0.730 | 0.764 | 0.749 | 0.704 | 0.809 | 0.771 | — | 0.789 | 0.843 |
| tools | NI-SSMBA | 0.792 | **0.854** | 0.786 | 0.707 | 0.746 | 0.875 | **0.888** | 0.792 | — | 0.866 |
| | NI-EDA | **0.859** | 0.846 | **0.834** | **0.847** | **0.790** | 0.837 | 0.886 | **0.819** | — | **0.894** |
| | NI-BT | 0.769 | 0.802 | 0.763 | 0.709 | 0.750 | 0.832 | 0.824 | 0.744 | — | 0.856 |
| | NI-RandRep | 0.792 | 0.825 | 0.816 | 0.717 | 0.724 | **0.885** | 0.877 | 0.801 | — | 0.845 |
| | ATC-NE | 0.745 | 0.789 | 0.775 | 0.744 | 0.632 | 0.698 | 0.856 | 0.737 | — | 0.799 |
| | ATC-MC | 0.749 | 0.798 | 0.772 | 0.745 | 0.634 | 0.737 | 0.876 | 0.777 | — | 0.793 |
| toys | NI-SSMBA | 0.779 | **0.830** | **0.813** | 0.766 | 0.695 | 0.808 | **0.872** | 0.872 | **0.891** | — |
| | NI-EDA | 0.696 | 0.790 | 0.805 | **0.821** | **0.790** | 0.837 | 0.834 | 0.860 | 0.840 | — |
| | NI-BT | 0.748 | 0.762 | 0.746 | 0.755 | 0.707 | **0.840** | 0.809 | 0.773 | 0.811 | — |
| | NI-RandRep | **0.813** | 0.788 | 0.783 | 0.770 | 0.660 | 0.788 | 0.838 | **0.873** | **0.891** | — |
| | ATC-NE | 0.656 | 0.469 | 0.743 | 0.550 | 0.628 | 0.515 | 0.792 | 0.703 | 0.771 | — |
| | ATC-MC | 0.694 | 0.482 | 0.726 | 0.539 | 0.651 | 0.571 | 0.805 | 0.730 | 0.777 | — |

Table 20: Full macro $\tau$ metrics for all pairs of training and test domains for BERT models trained on AWS.

| Train Domain | Measure | Test Domain | | | | | | | | | |
|---|---|---|---|---|---|---|---|---|---|---|---|
| | | slate | verbatim | facetoface | oup | nineeleven | fiction | telephone | travel | letters | government |
| slate | NI-SSMBA | — | 0.483 | 0.675 | 0.553 | 0.608 | 0.650 | 0.661 | 0.570 | 0.622 | 0.641 |
| | NI-EDA | — | 0.564 | 0.675 | 0.563 | 0.599 | 0.658 | 0.672 | 0.557 | 0.599 | 0.649 |
| | NI-BT | — | **0.669** | **0.742** | **0.736** | **0.756** | **0.727** | **0.754** | **0.751** | **0.738** | **0.791** |
| | NI-RandRep | — | 0.393 | 0.526 | 0.529 | 0.480 | 0.398 | 0.438 | 0.445 | 0.640 | 0.531 |
| | ATC-NE | — | 0.650 | 0.632 | 0.577 | 0.594 | 0.683 | 0.625 | 0.602 | 0.621 | 0.699 |
| | ATC-MC | — | 0.652 | 0.632 | 0.578 | 0.594 | 0.681 | 0.625 | 0.603 | 0.622 | 0.697 |
| fiction | NI-SSMBA | 0.577 | 0.537 | 0.692 | 0.482 | 0.539 | — | 0.629 | 0.520 | 0.546 | 0.598 |
| | NI-EDA | 0.632 | 0.584 | 0.691 | 0.608 | 0.557 | — | 0.631 | 0.529 | 0.540 | 0.608 |
| | NI-BT | **0.783** | **0.704** | **0.771** | **0.693** | **0.671** | — | **0.776** | **0.680** | **0.707** | **0.693** |
| | NI-RandRep | 0.495 | 0.446 | 0.522 | 0.531 | 0.437 | — | 0.434 | 0.464 | 0.531 | 0.493 |
| | ATC-NE | 0.538 | 0.461 | 0.630 | 0.428 | 0.513 | — | 0.542 | 0.469 | 0.581 | 0.536 |
| | ATC-MC | 0.530 | 0.450 | 0.631 | 0.427 | 0.513 | — | 0.543 | 0.463 | 0.576 | 0.534 |
| telephone | NI-SSMBA | 0.567 | 0.503 | **0.736** | 0.493 | 0.507 | 0.608 | — | 0.492 | 0.572 | 0.496 |
| | NI-EDA | 0.637 | 0.633 | 0.726 | 0.537 | 0.615 | **0.735** | — | 0.579 | **0.642** | 0.580 |
| | NI-BT | **0.657** | **0.668** | 0.722 | **0.637** | **0.663** | 0.639 | — | **0.621** | 0.602 | **0.614** |
| | NI-RandRep | 0.504 | 0.470 | 0.573 | 0.470 | 0.510 | 0.480 | — | 0.482 | 0.576 | 0.440 |
| | ATC-NE | 0.533 | 0.507 | 0.681 | 0.526 | 0.500 | 0.632 | — | 0.453 | 0.564 | 0.491 |
| | ATC-MC | 0.534 | 0.507 | 0.681 | 0.522 | 0.501 | 0.637 | — | 0.455 | 0.564 | 0.494 |
| travel | NI-SSMBA | 0.524 | 0.528 | 0.585 | 0.462 | 0.517 | 0.537 | 0.569 | — | 0.617 | 0.521 |
| | NI-EDA | 0.492 | 0.484 | 0.523 | 0.461 | 0.418 | 0.550 | 0.524 | — | 0.507 | 0.458 |
| | NI-BT | **0.619** | **0.634** | **0.630** | **0.660** | **0.681** | **0.655** | **0.645** | — | **0.713** | **0.699** |
| | NI-RandRep | 0.465 | 0.476 | 0.467 | 0.514 | 0.492 | 0.472 | 0.526 | — | 0.569 | 0.497 |
| | ATC-NE | 0.579 | 0.528 | 0.533 | 0.643 | 0.541 | 0.571 | 0.591 | — | 0.572 | 0.597 |
| | ATC-MC | 0.576 | 0.528 | 0.534 | 0.642 | 0.536 | 0.573 | 0.594 | — | 0.570 | 0.602 |
| government | NI-SSMBA | 0.578 | 0.531 | 0.600 | 0.568 | 0.565 | 0.617 | 0.589 | 0.563 | 0.616 | — |
| | NI-EDA | 0.581 | 0.582 | 0.565 | 0.535 | 0.594 | 0.624 | 0.519 | 0.552 | 0.566 | — |
| | NI-BT | **0.719** | **0.693** | **0.686** | **0.687** | **0.755** | **0.701** | **0.651** | **0.712** | **0.719** | — |
| | NI-RandRep | 0.274 | 0.268 | 0.357 | 0.444 | 0.464 | 0.301 | 0.247 | 0.395 | 0.452 | — |
| | ATC-NE | 0.624 | 0.464 | 0.581 | 0.666 | 0.440 | 0.601 | 0.616 | 0.524 | 0.641 | — |
| | ATC-MC | 0.621 | 0.465 | 0.583 | 0.667 | 0.446 | 0.602 | 0.616 | 0.522 | 0.642 | — |

Table 21: Full macro $\tau$ metrics for all pairs of training and test domains for CNN models trained on MNLI.

| Train Domain | Measure | Test Domain | | | | | | | | | |
|---|---|---|---|---|---|---|---|---|---|---|---|
| | | slate | verbatim | facetoface | oup | nineeleven | fiction | telephone | travel | letters | government |
| slate | NI-SSMBA | — | **0.743** | 0.746 | 0.709 | **0.767** | 0.742 | 0.727 | **0.771** | **0.782** | **0.819** |
| | NI-EDA | — | 0.580 | 0.695 | 0.674 | 0.696 | 0.703 | 0.655 | 0.651 | 0.671 | 0.737 |
| | NI-BT | — | 0.669 | 0.742 | 0.736 | 0.756 | 0.727 | 0.754 | 0.751 | 0.738 | 0.791 |
| | NI-RandRep | — | 0.393 | 0.526 | 0.529 | 0.480 | 0.398 | 0.438 | 0.445 | 0.640 | 0.531 |
| | ATC-NE | — | 0.689 | **0.768** | 0.786 | 0.728 | 0.752 | 0.761 | 0.744 | 0.743 | 0.766 |
| | ATC-MC | — | 0.684 | 0.761 | **0.794** | 0.730 | **0.761** | **0.789** | 0.765 | 0.744 | 0.785 |
| fiction | NI-SSMBA | 0.755 | 0.673 | 0.699 | **0.726** | **0.729** | — | 0.769 | **0.806** | **0.770** | **0.727** |
| | NI-EDA | 0.609 | 0.505 | 0.649 | 0.659 | 0.532 | — | 0.608 | 0.554 | 0.624 | 0.629 |
| | NI-BT | **0.783** | **0.704** | **0.771** | 0.693 | 0.671 | — | **0.776** | 0.680 | 0.707 | 0.693 |
| | NI-RandRep | 0.495 | 0.446 | 0.522 | 0.531 | 0.437 | — | 0.434 | 0.464 | 0.531 | 0.493 |
| | ATC-NE | 0.618 | 0.377 | 0.707 | 0.478 | 0.711 | — | 0.596 | 0.558 | 0.658 | 0.711 |
| | ATC-MC | 0.631 | 0.422 | 0.702 | 0.513 | 0.707 | — | 0.603 | 0.549 | 0.670 | 0.706 |
| telephone | NI-SSMBA | **0.720** | **0.768** | **0.793** | **0.792** | 0.668 | 0.720 | — | **0.813** | **0.758** | **0.788** |
| | NI-EDA | 0.670 | 0.667 | 0.706 | 0.617 | **0.718** | **0.795** | — | 0.642 | 0.732 | 0.661 |
| | NI-BT | 0.657 | 0.668 | 0.722 | 0.637 | 0.663 | 0.639 | — | 0.621 | 0.602 | 0.614 |
| | NI-RandRep | 0.504 | 0.470 | 0.573 | 0.470 | 0.510 | 0.480 | — | 0.482 | 0.576 | 0.440 |
| | ATC-NE | 0.462 | 0.486 | 0.692 | 0.575 | 0.641 | 0.542 | — | 0.644 | 0.697 | 0.702 |
| | ATC-MC | 0.519 | 0.551 | 0.717 | 0.641 | 0.637 | 0.597 | — | 0.665 | 0.684 | 0.708 |
| travel | NI-SSMBA | **0.700** | **0.806** | **0.737** | **0.676** | **0.744** | **0.707** | **0.848** | — | **0.789** | **0.783** |
| | NI-EDA | 0.483 | 0.453 | 0.486 | 0.493 | 0.489 | 0.515 | 0.483 | — | 0.522 | 0.495 |
| | NI-BT | 0.619 | 0.634 | 0.630 | 0.660 | 0.681 | 0.655 | 0.645 | — | 0.713 | 0.699 |
| | NI-RandRep | 0.465 | 0.476 | 0.467 | 0.514 | 0.492 | 0.472 | 0.526 | — | 0.569 | 0.497 |
| | ATC-NE | 0.155 | 0.320 | 0.234 | 0.559 | 0.533 | 0.146 | 0.360 | — | 0.633 | 0.630 |
| | ATC-MC | 0.175 | 0.370 | 0.320 | 0.613 | 0.576 | 0.219 | 0.443 | — | 0.648 | 0.646 |
| government | NI-SSMBA | **0.769** | **0.777** | **0.727** | 0.652 | 0.640 | **0.783** | **0.784** | **0.805** | **0.754** | — |
| | NI-EDA | 0.521 | 0.513 | 0.480 | 0.578 | 0.661 | 0.657 | 0.494 | 0.565 | 0.598 | — |
| | NI-BT | 0.719 | 0.693 | 0.686 | **0.687** | **0.755** | 0.701 | 0.651 | 0.712 | 0.719 | — |
| | NI-RandRep | 0.274 | 0.268 | 0.357 | 0.444 | 0.464 | 0.301 | 0.247 | 0.395 | 0.452 | — |
| | ATC-NE | 0.377 | 0.176 | 0.472 | 0.660 | 0.558 | 0.239 | 0.497 | 0.302 | 0.638 | — |
| | ATC-MC | 0.427 | 0.231 | 0.487 | 0.653 | 0.600 | 0.275 | 0.496 | 0.323 | 0.629 | — |

Table 22: Full macro $\tau$ metrics for all pairs of training and test domains for BERT models trained on MNLI.

| Model | Measure | Test Domain | | | | | | | | | |
|---|---|---|---|---|---|---|---|---|---|---|---|
| | | books | clothing | home | kindle | movies | pets | sports | tech | tools | toys |
| CNN | NI-SSMBA | **0.677** | 0.688 | **0.758** | **0.641** | **0.691** | **0.786** | **0.784** | **0.758** | 0.722 | 0.683 |
| | NI-EDA | 0.637 | 0.672 | 0.704 | 0.612 | 0.658 | 0.746 | 0.739 | 0.680 | 0.703 | 0.711 |
| | NI-BT | 0.601 | 0.532 | 0.572 | 0.509 | 0.521 | 0.509 | 0.587 | 0.536 | 0.571 | 0.529 |
| | NI-RandRep | 0.485 | 0.542 | 0.501 | 0.522 | 0.440 | 0.491 | 0.650 | 0.477 | 0.507 | 0.550 |
| | ATC-NE | 0.406 | **0.727** | 0.728 | 0.362 | 0.499 | 0.722 | 0.726 | 0.679 | **0.730** | **0.752** |
| | ATC-MC | 0.410 | 0.726 | 0.729 | 0.359 | 0.500 | 0.724 | 0.727 | 0.681 | **0.730** | **0.752** |
| BERT | NI-SSMBA | 0.790 | **0.850** | 0.868 | 0.664 | **0.899** | **0.864** | **0.855** | **0.874** | **0.880** | 0.845 |
| | NI-EDA | 0.739 | 0.848 | 0.825 | **0.731** | 0.793 | 0.836 | 0.825 | 0.828 | 0.837 | 0.841 |
| | NI-BT | 0.702 | 0.845 | 0.830 | 0.668 | 0.792 | 0.839 | 0.843 | 0.813 | 0.819 | **0.858** |
| | NI-RandRep | **0.839** | 0.841 | **0.891** | 0.694 | 0.864 | 0.834 | 0.747 | 0.825 | 0.850 | 0.831 |
| | ATC-NE | 0.613 | 0.721 | 0.712 | 0.618 | 0.695 | 0.666 | 0.733 | 0.693 | 0.708 | 0.755 |
| | ATC-MC | 0.621 | 0.735 | 0.730 | 0.616 | 0.711 | 0.691 | 0.749 | 0.710 | 0.726 | 0.764 |

Table 23: Full micro $\tau$ metrics for all test domains for CNN and BERT models trained on AWS.

| Model | Measure | Test Domain | | | | | | | | | |
|---|---|---|---|---|---|---|---|---|---|---|---|
| | | slate | verbatim | facetoface | oup | nineeleven | fiction | telephone | travel | letters | government |
| CNN | NI-SSMBA | 0.540 | 0.493 | 0.562 | 0.493 | 0.527 | 0.431 | 0.607 | 0.544 | 0.579 | 0.563 |
| | NI-EDA | 0.560 | 0.555 | 0.427 | 0.520 | 0.488 | 0.435 | 0.499 | 0.552 | 0.532 | 0.542 |
| | NI-BT | **0.698** | **0.684** | **0.611** | **0.713** | **0.713** | **0.595** | **0.676** | **0.714** | **0.697** | **0.722** |
| | NI-RandRep | 0.448 | 0.417 | 0.427 | 0.503 | 0.472 | 0.324 | 0.395 | 0.450 | 0.550 | 0.509 |
| | ATC-NE | 0.461 | 0.413 | 0.549 | 0.427 | 0.350 | 0.517 | 0.531 | 0.307 | 0.497 | 0.404 |
| | ATC-MC | 0.460 | 0.411 | 0.550 | 0.429 | 0.351 | 0.517 | 0.532 | 0.307 | 0.496 | 0.404 |
| BERT | NI-SSMBA | **0.782** | **0.710** | **0.742** | 0.652 | **0.724** | **0.756** | **0.779** | 0.691 | **0.768** | 0.700 |
| | NI-EDA | 0.563 | 0.556 | 0.481 | 0.622 | 0.586 | 0.545 | 0.529 | 0.607 | 0.607 | 0.630 |
| | NI-BT | 0.698 | 0.684 | 0.611 | **0.713** | 0.713 | 0.595 | 0.676 | **0.714** | 0.697 | **0.722** |
| | NI-RandRep | 0.448 | 0.417 | 0.427 | 0.503 | 0.472 | 0.324 | 0.395 | 0.450 | 0.550 | 0.509 |
| | ATC-NE | 0.375 | 0.389 | 0.660 | 0.606 | 0.593 | 0.414 | 0.523 | 0.511 | 0.655 | 0.687 |
| | ATC-MC | 0.405 | 0.434 | 0.676 | 0.632 | 0.614 | 0.466 | 0.554 | 0.530 | 0.663 | 0.693 |

Table 24: Full micro $\tau$ metrics for all test domains for CNN and BERT models trained on MNLI.

| Measure | Train Domain (Model) | | | | |
|---|---|---|---|---|---|
| | SVHN (NiN) | CIFAR10 (NiN) | CIFAR10 (ResNet) | CIFAR10 (VGG) | CINIC10 (CNN) |
| NI-RandAug | 0.733 | **0.797** | 0.746 | **0.869** | **0.759** |
| NI-Translate | **0.881** | 0.782 | **0.833** | 0.699 | 0.329 |
| NI-Erase | 0.324 | 0.409 | 0.708 | -0.183 | 0.606 |
| NI-FC | -0.033 | -0.015 | 0.568 | 0.628 | 0.289 |
| ATC-NE | 0.859 | 0.648 | 0.757 | 0.773 | 0.548 |
| ATC-MC | 0.843 | 0.605 | 0.756 | 0.767 | 0.553 |

Table 25: Full in-domain $\tau$ metrics for all test domains for image classification models and datasets.

| Model | Measure | Train Domain | | | | | | | | | |
|---|---|---|---|---|---|---|---|---|---|---|---|
| | | books | clothing | home | kindle | movies | pets | sports | tech | tools | toys |
| CNN | NI-SSMBA | 0.646 | **0.613** | **0.643** | 0.577 | 0.597 | 0.591 | 0.703 | 0.648 | 0.643 | 0.620 |
| | NI-EDA | **0.653** | 0.558 | 0.615 | **0.660** | 0.447 | **0.666** | **0.708** | 0.646 | **0.663** | 0.565 |
| | NI-BT | 0.569 | 0.449 | 0.582 | 0.537 | **0.626** | 0.605 | 0.568 | **0.653** | 0.640 | **0.652** |
| | NI-RandRep | 0.560 | 0.557 | 0.559 | 0.559 | 0.593 | 0.596 | 0.572 | 0.545 | 0.566 | 0.615 |
| | ATC-NE | 0.575 | 0.413 | 0.561 | 0.478 | 0.294 | 0.464 | 0.539 | 0.561 | 0.446 | 0.341 |
| | ATC-MC | 0.566 | 0.405 | 0.558 | 0.478 | 0.300 | 0.461 | 0.540 | 0.569 | 0.451 | 0.338 |
| BERT | NI-SSMBA | 0.735 | **0.874** | 0.868 | 0.693 | 0.780 | 0.871 | **0.884** | 0.851 | 0.851 | **0.884** |
| | NI-EDA | **0.875** | 0.788 | 0.814 | **0.700** | **0.870** | **0.894** | 0.818 | **0.858** | **0.882** | 0.810 |
| | NI-BT | 0.801 | 0.811 | 0.857 | 0.638 | 0.835 | 0.813 | 0.860 | 0.846 | 0.803 | 0.750 |
| | NI-RandRep | 0.839 | 0.841 | **0.891** | 0.694 | 0.864 | 0.834 | 0.747 | 0.825 | 0.850 | 0.831 |
| | ATC-NE | 0.733 | 0.724 | 0.822 | 0.640 | 0.721 | 0.711 | 0.778 | 0.783 | 0.799 | 0.783 |
| | ATC-MC | 0.742 | 0.746 | 0.813 | 0.630 | 0.730 | 0.725 | 0.814 | 0.780 | 0.774 | 0.739 |

Table 26: Full in-domain $\tau$ metrics for all test domains for CNN and BERT models trained on AWS.

| Model | Measure | Train Domain | | | | |
|-------|---------|-------|---------|-----------|--------|------------|
| | | slate | fiction | telephone | travel | government |
| CNN | NI-SSMBA | 0.664 | 0.655 | 0.753 | 0.692 | 0.758 |
| | NI-EDA | 0.629 | 0.723 | **0.786** | 0.652 | 0.755 |
| | NI-BT | **0.751** | **0.765** | 0.637 | **0.786** | **0.826** |
| | NI-RandRep | 0.437 | 0.471 | 0.546 | 0.570 | 0.491 |
| | ATC-NE | 0.704 | 0.662 | 0.719 | 0.715 | 0.722 |
| | ATC-MC | 0.703 | 0.660 | 0.722 | 0.716 | 0.728 |
| BERT | NI-SSMBA | 0.713 | 0.754 | 0.766 | 0.785 | **0.839** |
| | NI-EDA | 0.608 | 0.664 | 0.781 | 0.575 | 0.726 |
| | NI-BT | **0.751** | 0.765 | 0.637 | **0.786** | 0.826 |
| | NI-RandRep | 0.437 | 0.471 | 0.546 | 0.570 | 0.491 |
| | ATC-NE | 0.722 | **0.796** | 0.740 | 0.737 | 0.701 |
| | ATC-MC | 0.745 | 0.791 | **0.791** | 0.719 | 0.696 |

Table 27: Full in-domain $\tau$ metrics for all train domains for CNN and BERT models trained on MNLI.

| Measure | Domain Shifts | | ImageNet-A | | |
|---|---|---|---|---|---|
| | $R^2$ | Macro $\tau$ | $R^2$ | Macro $\tau$ | ID $\tau$ |
| NI-RandAug | 0.091 | 0.061 | — | — | — |
| NI-Translate | 0.159 | 0.099 | — | — | — |
| NI-Erase | 0.174 | 0.153 | — | — | — |
| NI-FC | 0.139 | 0.095 | — | — | — |
| ATC-NE | 0.272 | 0.136 | — | — | — |
| ATC-MC | 0.279 | 0.135 | — | — | — |

(a) Standard deviations for ImageNet scale results. No standard deviations are reported for ImageNet-A or ID $\tau$ since we report only a single value.

| Measure | CI10 | | | Numbers | | |
|---|---|---|---|---|---|---|
| | $R^2$ | Macro $\tau$ | ID $\tau$ | $R^2$ | Macro $\tau$ | ID $\tau$ |
| NI-RandAug | 0.044 | 0.054 | 0.048 | 0.021 | 0.026 | — |
| NI-Translate | 0.170 | 0.181 | 0.197 | 0.043 | 0.032 | — |
| NI-Erase | 0.254 | 0.348 | 0.345 | 0.065 | 0.033 | — |
| NI-FC | 0.239 | 0.225 | 0.255 | 0.074 | 0.083 | — |
| ATC-NE | 0.169 | 0.115 | 0.091 | 0.109 | 0.059 | — |
| ATC-MC | 0.168 | 0.101 | 0.093 | 0.077 | 0.035 | — |

(b) Standard deviations for small scale image results. No standard deviation is reported for Numbers ID $\tau$ since we report only a single value.

| Measure | CNN | | | | RoBERTa | | | |
|---|---|---|---|---|---|---|---|---|
| | $R^2$ | Macro $\tau$ | Micro $\tau$ | ID $\tau$ | $R^2$ | Macro $\tau$ | Micro $\tau$ | ID $\tau$ |
| NI-SSMBA | 0.079 | 0.067 | 0.046 | 0.035 | 0.012 | 0.063 | 0.064 | 0.065 |
| NI-EDA | 0.108 | 0.051 | 0.041 | 0.072 | 0.016 | 0.058 | 0.040 | 0.055 |
| NI-BT | 0.161 | 0.078 | 0.032 | 0.060 | 0.021 | 0.060 | 0.061 | 0.063 |
| NI-RandRep | 0.069 | 0.080 | 0.054 | 0.021 | 0.015 | 0.060 | 0.055 | 0.055 |
| ATC-NE | 0.126 | 0.110 | 0.143 | 0.092 | 0.168 | 0.135 | 0.044 | 0.051 |
| ATC-MC | 0.125 | 0.109 | 0.143 | 0.091 | 0.139 | 0.112 | 0.048 | 0.050 |

(c) Standard deviations on sentiment analysis results.

| Measure | CNN | | | | RoBERTa | | | |
|---|---|---|---|---|---|---|---|---|
| | $R^2$ | Macro $\tau$ | Micro $\tau$ | ID $\tau$ | $R^2$ | Macro $\tau$ | Micro $\tau$ | ID $\tau$ |
| NI-SSMBA | 0.098 | 0.060 | 0.048 | 0.043 | 0.035 | 0.046 | 0.040 | 0.041 |
| NI-EDA | 0.107 | 0.068 | 0.046 | 0.060 | 0.108 | 0.087 | 0.044 | 0.075 |
| NI-BT | 0.045 | 0.049 | 0.042 | 0.063 | 0.045 | 0.049 | 0.042 | 0.063 |
| NI-RandRep | 0.196 | 0.079 | 0.061 | 0.049 | 0.196 | 0.079 | 0.061 | 0.049 |
| ATC-NE | 0.108 | 0.068 | 0.076 | 0.022 | 0.206 | 0.180 | 0.111 | 0.032 |
| ATC-MC | 0.108 | 0.068 | 0.076 | 0.024 | 0.174 | 0.164 | 0.100 | 0.038 |

(d) Standard deviations on NLI results.

Table 28: Standard deviations for reported values in Table 2

