# OpenReview forum: "Predicting Out-of-Domain Generalization with Neighborhood Invariance"
_TMLR — Accepted by TMLR_

### Review · Reviewer_qQw6 · 2023-04-01

**Summary Of Contributions:**

The paper investigates the correlation between neighborhood invariance and generalization capacity. The proposed neighborhood invariance measure is a simple and effective way to calculate a classifier's output invariance in a local transformation neighborhood, making it applicable even in out-of-domain settings, where other methods cannot be used.


**Audience:**

Yes

**Claims And Evidence:**

No

**Requested Changes:**

Please read the above weaknesses. They are critical to secure my recommendation for acceptance.

**Strengths And Weaknesses:**

Strengths:

- The proposed method is easy, compelling, and seems effective without making extra assumptions as opposed to the methods in the literature.
- The paper is well-written and easy to follow.
- The paper addresses an important issue in machine learning and provides an extensive set of experiments.

Weaknesses:
I have a few concerns in the experimental section:

1. The vision datasets used in the experiments are extremely small in pixels (32x32). Even CINIC that includes some images from ImageNet downsize them to 32x32. Since the main contribution of the paper comes from data transformations it would be more informative and conclusive to see results on more realistic datasets such as ImageNet and then generalizing to ImageNet variants which are standard DG testbeds.
2. Another note on my previous point, although results presented in Figure 6 (by the way, it should be titled as Table 6?) are shown to support the claim that "training models with data augmentations doesn't interfere with the outcomes of the proposed method and the method is still effective", I am curious to see results shown in Table 6 for a dataset with better resolution (eg. ImageNet but I am open to other datasets too)
3. I am curious to know why authors use CNN models for NLP tasks given that CNNs in NLP are neither the common practice nor the state of the art. Also why not having a transformer based models for vision experiments?
4. Another missing point I think is to see the effect of pretraining on the effectiveness of the proposed method. Pretraining is believed to increase generalizability so is the proposed method still effective under circumstances where DG is stronger? Especially contrastive pertaining methods in vision which leverage lots of augmentation techniques.
5. Last but not least, it would be very helpful to see the standard deviations of the numbers in your paper to get a better sense of how spread out they are. Sometimes they are too close which makes it hard to assess.

---

> ### Author Response · Authors · 2023-04-13
> **Author Response**
>
> Thank you for your review. We respond to specific concerns below.
>
> ***
> **The vision datasets used in the experiments are extremely small in pixels (32x32)**
>
> We respond to these concerns in the general response (Regarding the limited model and dataset scale).
>
> ***
> **I am curious to see results shown in Table 6 for a dataset with better resolution (eg. ImageNet but I am open to other datasets too)**
>
> This is an interesting point. We will add the results on models trained with augmentations from the Imagenet-Testbed mentioned in the general response.
>
> ***
> **I am curious to know why authors use CNN models for NLP tasks given that CNNs in NLP are neither the common practice nor the state of the art. Also why not having a transformer based models for vision experiments?**
>
> We respond to some of these concerns in our general response (Regarding the novelty of our work, Regarding the limited model and dataset scale).
>
> Although CNNs are not state of the art, in our experiments they perform better than RNNs at certain tasks such as sentiment classification [1]. In addition, we note that smaller scale RNN and CNN models are still useful in domain specific contexts for which fast inference speed is required and pretrained knowledge unnecessary.
>
> ***
> **Another missing point I think is to see the effect of pretraining on the effectiveness of the proposed method.**
>
> This is an interesting point. As mentioned in our general response, we plan to include large pretrained vision transformers as well as linear probes on CLIP models in our additional experiments.
>
> ***
> **it would be very helpful to see the standard deviations of the numbers in your paper to get a better sense of how spread out they are**
>
> We will add an additional table with standard deviations to the appendix. Unfortunately including these in the main tables is difficult due to the size constraints.
> ***
> [1] Yoon Kim. 2014. Convolutional neural networks for sentence classification.

---

### Review · Reviewer_yuPF · 2023-04-02

**Summary Of Contributions:**

In this paper, the authors propose a new measure, namely *neighborhood invariance*, to characterize out-of-domain generalization. It is implemented by first selecting a set of data transformations and then calculating the percentage of points classified into the most commonly predicted class in the neighborhood of the input test point.  Extensive experimental results show the proposed measure is consistent with the out-of-domain generalization.

**Audience:**

Yes

**Broader Impact Concerns:**

The authors did not discuss the broad impact of their method in the paper. I also do not see any obvious ethical implications of this work in the near future.

**Claims And Evidence:**

No

**Requested Changes:**

See **Weaknesses** above.

**Strengths And Weaknesses:**

**Strengths**:
+ The idea is simple but intuitively reasonable.
+ The implementation is straightforward.
+ The paper is generally well written and easy to follow.

**Weaknesses:**

 - My main concern is how to select a set of **appropriate** data transformations. In my opinion, it is the key to the proposed measure. The appropriateness entirely determines the validity of the measure. Unfortunately, it seems that the authors skip this significant point in the paper. At least, the authors should provide some general rules to guide how to select them in different tasks.
- How to guarantee the selected data transformations are still well defined? That is, applying the transformation will not lead to an undefined label. This is another key technical point, but not well discussed in the paper.
- In light of the two issues above, it is hard to justify the extent to which the proposed measure works in practice.
- In the section of related work, the authors missed a large body of work on invariance in the causality community.
- In the image classification task, the types of data transformations considered in the experiments are too simple to cover many real-world scenarios. I suggest the authors conduct experiments on more real-world datasets, e.g., DomainBed (https://github.com/facebookresearch/DomainBed).
- The description of the experimental setup, eps the sections of Data, and Model and Hyperparameter Space, are too vague, which should be more detailedly explained.
- The baseline methods are too simple to demonstrate the proposed measure. The authors should compare with those in the community of adversarial robustness.
- Some typos: 1) pp2: "our measure can applied" -> "our measure can be applied"? 2) pp3:"then motivate our formulation"?

---

> ### Author Response · Authors · 2023-04-13
> **Author Response**
>
> Thank you for your review. We have fixed the typos mentioned and respond to specific concerns below.
>
> ***
> **My main concern is how to select a set of appropriate data transformations…How to guarantee the selected data transformations are still well defined? That is, applying the transformation will not lead to an undefined label.**
>
> We respond to these concerns in the general response (Regarding the selection of appropriate data transformations).
>
> ***
> **In the section of related work, the authors missed a large body of work on invariance in the causality community.**
>
> Thank you for the suggestion. We plan to add additional related work from these communities in our related work. We discuss some of the differences of our work with those in the invariant learning community in our general response (Regarding the novelty of our work).
>
> ***
> **In the image classification task, the types of data transformations considered in the experiments are too simple to cover many real-world scenarios. I suggest the authors conduct experiments on more real-world datasets, e.g., DomainBed (https://github.com/facebookresearch/DomainBed)**
>
> We note that data transformations are not meant to cover real world shifts themselves, but rather to characterize the **local** invariance of a given model on a specific test point. We provide additional clarification on the contributions of our paper in the general response (Regarding the novelty of our work).
>
> We agree that larger scale real world datasets should be considered in our experiments. Accordingly we plan to add additional results on ImageNet scale datasets with real domain shifts on a large suite of models. We detail these experiments in the general response (Regarding the limited model and dataset scale).
>
> ***
> **The description of the experimental setup, eps the sections of Data, and Model and Hyperparameter Space, are too vague, which should be more detailedly explained.**
>
> The specifics of our experimental setup are provided in greater detail in Appendix A, describing our data preprocessing steps, model architecture choices along with specific hyperparameter choices, model training and early stopping criteria, as well as hyperparameters for all the transformations considered. Additional details for pretrained vision models can be found in the respective papers [1, 2]. If there are specific details that are missing from the Appendix please let us know and we will be happy to add them in.
>
> ***
> **The baseline methods are too simple to demonstrate the proposed measure. The authors should compare with those in the community of adversarial robustness.**
>
> Models trained with adversarial robustness are included in the Imagenet-Testbed [3], so our additional results will include these models. We also plan to evaluate all models in the testbed on ImageNet-A, a dataset of adversarially selected images on which many models perform extremely poorly.
>
> An interesting experiment to consider would be to measure whether our measure correlates with robustness to model specific adversarial attacks. However, since most adversarial attacks require access to model gradients or the training set and we assume only access to a black box classifier without detailed gradient or weight information, measuring robustness to these attacks is outside of the scope of our work.
> ***
>
> [1] Yiding Jiang, Pierre Foret, Scott Yak, Daniel M. Roy, Hossein Mobahi, Gintare Karolina Dziugaite, Samy Bengio, Suriya Gunasekar, Isabelle Guyon, and Behnam Neyshabur. NeurIPS 2020 Competition: Predicting Generalization in Deep Learning.
>
> [2] Yiding Jiang, Dilip Krishnan, Hossein Mobahi, and Samy Bengio. Predicting the generalization gap in deep networks with margin distributions.
>
> [3] Taori et al. Measuring Robustness to Natural Distribution Shifts in Image Classification. NeurIPS 2020.

---

### Review · Reviewer_8wmA · 2023-04-02

**Summary Of Contributions:**

The authors introduce "neighborhood invariance," a simple and versatile measure to evaluate the generalization capacity of machine learning models, particularly in out-of-domain (OOD) settings. This measure calculates the proportion of points classified into the most common predicted class within a transformation neighborhood. Unlike existing methods, it does not rely on strong assumptions or test point labels. The paper evaluates the measure on various tasks, using over 4,000 models and 100 train/test domain pairs. The neighborhood invariance measure consistently outperforms baselines and correlates strongly with actual generalization. Despite limitations related to data transformations and extreme OOD settings, the measure provides an alternative approach to characterizing and comparing model generalization across different experimental settings.

**Audience:**

No

**Broader Impact Concerns:**

I don't think there are any ethical concerns for this paper.

**Claims And Evidence:**

Yes

**Requested Changes:**

## Scaling up models and datasets
- For image classification, it is suggested that the authors follow prior works [1,2,3,5,6] and evaluate modern pre-trained networks (e.g., DenseNet, ViT, CLIP) on modern distribution shifts datasets (ImageNetV2/ObjectNet/ImageNet-A/ImageNet-R/ImageNet-C/WILDS-IWildCam/WILDS-FMoW, etc.).

- For text evaluation, the authors should consider following BIG-Bench [4] to evaluate modern models (e.g., GPT-2, GPT-Neo, T5, FlanT5, LLaMA) on up-to-date text OOD datasets (such as those in BIG-Bench).

## Discussion of Related Works
- It is recommended to add a discussion of previous distribution shift works on benchmark curation or large-scale empirical studies, such as [1,2,3,4,5,6].

[1] Miller et al. Accuracy on the Line: On the Strong Correlation Between Out-of-Distribution and In-Distribution Generalization. ICML 2021.

[2] Wortsman et al. Robust fine-tuning of zero-shot models. CVPR 2022.

[3] Baek et al. Agreement-on-the-line: Predicting the Performance of Neural Networks under Distribution Shift. NeurIPS 2022.

[4] Beyond the Imitation Game: Quantifying and extrapolating the capabilities of language models.

[5] Koh et al. WILDS: A Benchmark of in-the-Wild Distribution Shifts. ICML 2021.

[6] Gulrajani et al. In Search of Lost Domain Generalization. ICLR 2021.


**Strengths And Weaknesses:**

## Strengths

+ **Versatility**: The proposed neighborhood invariance measure is versatile, as it can be applied to various tasks (image classification, sentiment analysis, and natural language inference) and experimental settings. Unlike existing methods, the neighborhood invariance measure does not depend on the test point's true label and makes no assumptions about the data distribution or model.
+ **Strong correlation with generalization**: The measure consistently outperforms baselines and correlates strongly with actual generalization across tasks and experimental settings. The measure is particularly useful in OOD settings, where existing methods often struggle.

## Weaknesses

- **Reliance on appropriate data transformations:** The main limitation of this work is the reliance on selecting suitable data transformations. In certain tasks or domains, this might be challenging, and poor transformation choices can lead to inappropriately high estimates. A key challenge of real-world OOD shifts is their difficulty to characterize, as they often cannot be expressed through common data augmentations.

- **Limited Model & Dataset Scale:** The proposed evaluation metric is not surprising to researchers in invariant representation learning. Thus, the main contribution of this paper is its empirical evaluation. However, the models and datasets used in the study fall behind modern practice, making the paper less interesting or useful to the community. For image classification, the authors use CIFAR-10 variants and pre-trained networks up to ResNet, while current OOD robustness studies focus on larger pretrained networks and higher-resolution datasets. For text evaluations, the sentiment analysis (SA) and natural language inference (NLI) datasets also fall behind modern practices, as do the BERT family models. The study and evaluation results on small models and datasets may not be applicable to modern practices.

- **Insufficient discussion of related works:** The authors have missed distribution shift works in the related works discussion. It is recommended to add a paragraph discussing these works in the revision. See the section of requested changes below for details.

---

> ### Author Response · Authors · 2023-04-13
> **Author Response**
>
> Thank you for your review. We respond to specific concerns below.
>
> ***
> **The main limitation of this work is the reliance on selecting suitable data transformations. In certain tasks or domains, this might be challenging, and poor transformation choices can lead to inappropriately high estimates.**
>
> We respond to these concerns in the general response (Regarding the selection of appropriate data transformations)
>
> ***
> **The proposed evaluation metric is not surprising to researchers in invariant representation learning. Thus, the main contribution of this paper is its empirical evaluation.**
>
> We respond to these concerns in the general response (Regarding the novelty of our work).
>
> ***
> **For text evaluations, the sentiment analysis (SA) and natural language inference (NLI) datasets also fall behind modern practices, as do the BERT family models…the authors should consider following BIG-Bench [4] to evaluate modern models (e.g., GPT-2, GPT-Neo, T5, FlanT5, LLaMA) on up-to-date text OOD datasets (such as those in BIG-Bench).**
>
> We agree that it would be interesting to consider modern LLMs in our experimental settings. However, in order to ensure we are measuring generalization in a true OOD setting, we require that our models are trained only on the given ID dataset and do not observe the OOD dataset. Since most of these large models are already trained exactly on the data and tasks we use for benchmarks or have seen the examples in their datasets in the case of large language models, we cannot use them in our experimental setup.
>
> ***
> **The authors have missed distribution shift works in the related works discussion...It is recommended to add a discussion of previous distribution shift works on benchmark curation or large-scale empirical studies, such as [1,2,3,4,5,6].**
>
> Thank you for this suggestion. We will add these to our related works.
> ***

---

### Author Response · Authors · 2023-04-13
**General Response**

We thank the reviewers for their comments and are glad they found our method simple and compelling (8wmA, yuPF, qQw6), the problem we investigate important (qQw6), our set of experiments extensive (qQw6), and our manuscript well-written and easy to follow (yuPF, qQw6).

Below we respond to shared concerns by the reviewers and detail the changes we plan to make to the manuscript. We will update the reviewers as they are completed.

***
### **Regarding the limited model and dataset scale**

Reviewers suggested a large set of additional experiments covering ImageNet scale datasets and robustness benchmarks, larger modern model architectures including vision transformers, as well as models trained with adversarial robustness, data augmentation, and pretraining. Accordingly, we plan to run additional experiments on the Imagenet Testbed [1] that includes a wide range of models including Vision Transformers, CLIP embedding models, DenseNet, lResNeXt, EfficientNets, Dual path networks, and many more. We focus on the evaluation of these models on realistic dataset shifts from ImageNet to ImageNetV2, ObjectNet, ImageNet-Vid-Anchors, YTBB-Anchors, as well as the adversarial ImageNet-A. Since the models in this testbed include models trained with adversarial robustness, large amounts of additional data, data augmentations, and contrastive pretraining, we will be able to slice the results to answer many of the questions posed by reviewers.

In total, these experiments cover an additional 401 models evaluated on a set of 6 datasets for a full set of 2,406 more evaluations. This is in addition to the original set of experiments in our paper which covers 4,243 models trained and evaluated on over 100 train/test pairs for a full set of 110,072 evaluations. **This brings the total set of experiments in our work to about 4,644 models and 112,118 evaluations.**
Given the large scale of the experiments requested, we may not be able to complete them in the allotted review period. Note that our evaluations for a single model on a single dataset takes `nsamples * naugmentations = 10 * 5 = 50` times longer to run than evaluation on the original data. On average each individual evaluation thus far has taken approximately 4-8 hours to complete. This is made more difficult due to the unfortunate timing of a server migration of the Imagenet Testbed models and datasets which means some models and datasets are currently unavailable to access. **Although we are trying our best to work around the issues, we ask the reviewers and AE for their continued patience and consideration as we complete and update this large additional set of results.**

***
### **Regarding the novelty of our work**

We would like to clarify that the main contribution of our work is not to evaluate the generalization properties of state-of-the-art models, but rather to formalize a simple and widely applicable measure of generalization across many models and datasets, then characterize the behavior of our measure across many experimental settings.

Although it is well known that data augmentation and invariance to perturbations improves the generalization properties of models, this property is typically measured with respect to a test example’s true label [2] or the model’s unperturbed prediction [3]. In contrast, we measure invariance with respect to the local transformation neighborhood and not any specific predictions or labels. These local transformations are not large enough to shift domains, in contrast to the large body of work in the robust and invariant learning communities which consider explicit invariance to these domain shifts. In addition, our work makes fewer assumptions than these prior investigations into model robustness, requiring models to be trained with no knowledge or access to other domains, then evaluated in a black box setting without access to model weights, gradients, or training data.

***

---

> ### Author Response · Authors · 2023-04-13
>
> ***
> ### **Regarding the selection of appropriate data transformations**
>
> We present analysis on the choice of data transformations in Section 5.4 (Selecting Transformations). We show that our measure is relatively robust to the specific transformation selected as most transformations achieve similar correlations. We observe that transformations that cause a large increase in entropy are poor choices as hypothesized, since they cause model predictions to become much closer to random. Although some input data domains may not have a strong set of established data transformations, our ablations indicate that even simple transformations (such as a small translation in the case of images) can achieve strong correlations.
>
> In practice, in order to evaluate the appropriateness of a transformation, we can measure the change in entropy on the test set when applied and ensure it is not too large. We find in our experiments a maximum difference of 0.1 entropy is a good guideline for transformation selection. Even in the rare case that the entropy difference cannot be measured before utilizing our method, aggregating multiple transformations as in the case of RandAugment can help to ensure that even if a single poor transformation is selected, averaging with other transformations ensures our measure still works as expected.
> ***
> ### **Additional Related Work**
> We thank the reviewers for their suggestions for missing related work and plan to add the additional citations to our related works section.
> ***
>
> [1] Taori et al. Measuring Robustness to Natural Distribution Shifts in Image Classification. NeurIPS 2020.
>
> [2]  Hendrycks, D. and Dietterich, T. Benchmarking neural network robustness to common corruptions and perturbations. ICLR 2019.
>
> [3] Sumukh Aithal K, Dhruva Kashyap, and Natarajan Subramanyam. Robustness to Augmentations as a Generalization metric. Arxiv 2021.

---

### Author Response · Authors · 2023-04-21
**Updated Manuscript**

We would like to thank the reviewers and AE for their patience. We have updated the manuscript with the requested changes, which we detail specifically below:
- Additional experiments evaluating 196 models from the ImageNet Testbed [1]. These models cover various architectures including DenseNet, EfficientNet, ResNeXt, and Vision Transformers, as well as linear probes on pre-trained CLIP embeddings and more. We evaluate these models on 7 datasets: standard domain shifts of ImageNetV2, ImageNet-Sketch, ImageNet-R, ObjectNet, YTBB, and ImageNet-Vid, as well as an adversarial shift of Imagenet-A. We present results in Table 2a).
- We find that NI-RandAug outperforms ATC methods on standard ImageNet domain shifts. On adversarial data, where ATC methods fail completely, our NI based methods maintain strong performance. These results further support the conclusions drawn from the smaller scale experiments on CI10 and Numbers datasets.
- Additional ablations on the effects of robustness interventions (adversarial training, data augmentations, etc.) as well as additional data (contrastive pretraining, zero shot linear probes, etc.) by analyzing subsets of the models evaluated above. We find that, compared to standard training, additional robustness and data slightly degrade macro $\tau$ but increase $R^2$. However, compared to the performance on all models, NI-RandAug performs fairly consistently across all model subsets.
- Standard deviations for all results in the main Table 2 added to the Appendix in Table 28. We find that in general, ATC methods exhibit much higher variance compared to NI based methods. For example, on ImageNet datasets shifts, NI-RandAug exhibits an $R^2$ variance of 0.091, whereas ATC methods exhibit a variance of 0.27, indicating our method both performs better and is more consistent across datasets.
- Additional related work added on different notions of invariance as well as methods that analyze linear relationships between agreement, ID accuracy, and OOD accuracy.

We hope these updates help to answer the insightful questions and concerns brought up by reviewers.

[1] Taori et al. Measuring Robustness to Natural Distribution Shifts in Image Classification. NeurIPS 2020.

---

### Decision · Action_Editors · 2023-05-09

**Recommendation:** Accept as is

**Comment:**

The submission introduces neighborhood invariance, a measure of classifier output invariance to "local" transformations which is designed to characterize model generalization without making strong assumptions. Given a test example and some transformation neighborhood, the approach computes the proportion of transformations of the example classified as the most commonly predicted class within the neighborhood. The metric is then averaged over all examples of the test dataset.

Neighborhood invariance is evaluated on image classification, sentiment analysis, and natural language inference tasks, where it is shown to correlate well with OOD generalization.

Following the author-reviewer discussion, and after changes to the related work discussion and the incorporation of new experimental results, the reviewer consensus is that the submission is interesting to a wide audience and makes a valuable and well-supported empirical contribution.

**Audience:**

Reviewers note the submission's wide applicability (8wmA) and its writing quality (yuPF, qQw6).

**Claims And Evidence:**

All reviewers initially expressed concern in one way or another over the limited model and dataset scale in the experiments, but appreciate the author's efforts towards evaluating on larger-scale datasets such as ImageNet variants and feel that the revision addresses most of their concerns.